# NextlocLLM: Next Location Prediction Using LLMs

**SHUAI LIU, Ning Cao, Yile Chen, Yue Jiang, Gao Cong**
College of Computing and Data Science, Nanyang Technological University
50 Nanyang Avenue, Singapore, 639798
`{SHUAI004@e, Ning.Cao@, yile001@e,yue013@e,gaocong@}.ntu.edu.sg`

## Abstract

Next location prediction is a critical task in human mobility analysis and serves as a foundation for various downstream applications. Existing methods typically rely on discrete IDs to represent locations, which inherently overlook spatial relationships and cannot generalize across cities. In this paper, we propose NextLocLLM, which leverages the advantages of large language models (LLMs) in processing natural language descriptions and their strong generalization capabilities for next location prediction. Specifically, instead of using IDs, NextLocLLM encodes locations based on continuous spatial coordinates to better model spatial relationships. These coordinates are further normalized to enable robust cross-city generalization. Another highlight of NextlocLLM is its LLM-enhanced POI embeddings. It utilizes LLMs' ability to encode each POI category's natural language description into embeddings. These embeddings are then integrated via nonlinear projections to form this LLM-enhanced POI embeddings, effectively capturing locations' functional attributes. Furthermore, task and data prompt prefix, together with trajectory embeddings, are incorporated as input for partly-frozen LLM backbone. NextLocLLM further introduces prediction retrieval module to ensure structural consistency in prediction. Experiments show that NextLocLLM outperforms existing models in next location prediction, excelling in both supervised and zero-shot settings.

## 1 Introduction

With the rapid advancement of smart city infrastructure and positioning techniques, the acquisition of human mobility trajectories has become increasingly widespread, offering unprecedented research opportunities (Yabe et al., 2024a). Accurately predicting a user's next location holds significant value across multiple key domains. For urban planning and traffic management, forecasting mobility patterns can optimize traffic flow, reduce congestion, and improve the efficiency of public resource allocation (Medina-Salgado et al., 2022; Kraemer et al., 2020). In disease control, predicting population movements aids in tracking epidemic spread and formulating more effective prevention measures (Ceder, 2021). Moreover, accurate next location prediction is crucial for service providers who offer location-based recommendations and route planning, as it significantly enhances the quality of personalized services, delivering experiences that better meet users' needs (Lian et al., 2020).

Early methods for next location prediction mainly rely on manually designed features like behavioral sequences, user profiles, and temporal factors. Using these features, researchers employ statistical methods and time-series analysis to model and predict movement patterns (Noulas et al., 2012; Ying et al., 2014; Zhao et al., 2016). However, these methods heavily depend on feature engineering, which usually requires intensive efforts and exhibits limited performance. Later, neural network models like RNNs and Transformers are used to capture high-order transition patterns and dependencies (Feng et al., 2018; Yao et al., 2017; Kong & Wu, 2018). Some studies further integrate geospatial information by constructing graph neural networks to explore spatial correlations (Liu et al., 2016; Lian et al., 2020; Yao et al., 2023). These deep learning-based models typically learn embedding tables based on location IDs to represent locations. However, the learned embeddings inherently lack generalization capability, especially when applied to new locations from unseen cities or scenarios.

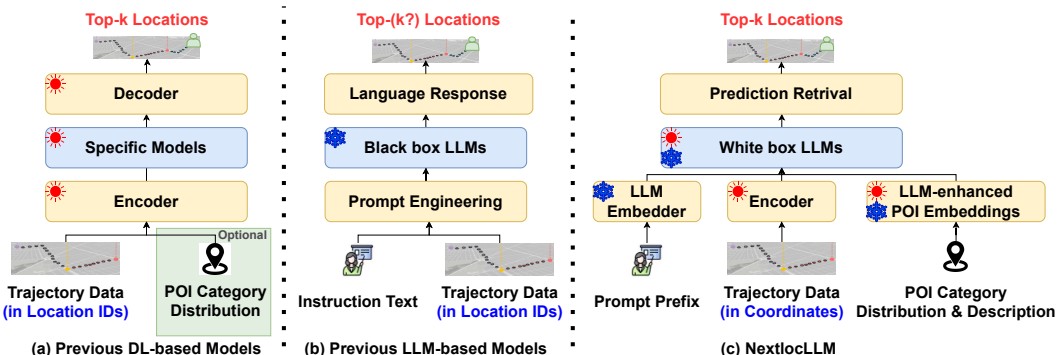

Figure 1: Comparison between deep-learning models, existing LLM-based models, and Nextlo-cLLM.White-box LLMs means models where their internal parameters are directly accessible. Black-box LLMs are typically accessed through APIs or encapsulated interfaces.

With the success of large language models (LLMs) (Achiam et al., 2023; Touvron et al., 2023b;a), researchers have started exploring the potential of using LLMs for next location prediction (Wang et al., 2023; Liang et al., 2024; Beneduce et al., 2024). However, these methods rely solely on utilizing prompts designed to describe trajectories denoted by location IDs, which introduces several limitations. First, these models use discrete location IDs, which fails to capture the geographic distances between locations, thus overlooking spatial relationships and reducing prediction accuracy (Liu et al., 2016). Second, the reliance on location IDs severely limits model transferability across cities, as the same ID in different cities usually represents completely different locations with varying features, making cross-city generalization infeasible (Jiang et al., 2021). Moreover, the distribution of point-of-interest (POI) categories, which provides functional information about locations, is not considered in these models due to the constraints of prompt length (Wang et al., 2023; Liang et al., 2024). Lastly, even when prompts are explicitly designed with clear output specifications, these models may still produce outputs that do not adhere to instructions (e.g., produce fewer location candidates than specified), posing challenges for subsequent structured analysis and application (see Appendix B).

To overcome these limitations, we introduce NextLocLLM, the first model that integrates LLMs in next location prediction structure, rather than relying solely on prompt-based approaches. Unlike traditional methods that depend on discrete location IDs, NextLocLLM leverages spatial coordinates to enhance its understanding of spatial relationships between locations. By normalizing coordinates, geographic information of different cities is mapped to a common numerical range, thus improving NextLocLLM's transferability and generalization across different urban environments. To better capture the functional characteristics of locations, NextLocLLM incorporates POI category distribution as a key feature. Leveraging the comprehensive power of LLMs, NextLocLLM generates semantic embeddings based on natural language descriptions of each POI category. These semantic embeddings are then weighted according to the frequency of each POI category at a given location, forming a weighted sum that represents the composite POI characteristics as the LLM-enhanced POI embeddings for the subsequent input. Additionally, prompt prefix is designed to provide LLM with rich contextual information, further enhancing its understanding of mobility data and prediction task. In output phase, the model employs a prediction retrieval module which employs KD-tree to convert output coordinates into the top-k most probable predicted locations, ensuring structured and clear outputs. Figure 1 illustrates the differences between NextLocLLM and existing deep learning or LLM-based next location prediction models. Extensive experiments demonstrate that NextLocLLM exhibits strong competitiveness in next location prediction tasks, excelling in both fully supervised or zero-shot scenarios. In summary, our main contributions are as follows:

- We propose NextLocLLM, the first model to integrate LLM in next location prediction structure, without solely using prompts. By utilizing normalized spatial coordinates to represent locations, NextLocLLM more accurately captures spatial relationships, as well as enhances its transferability and generalization in different urban environments.

- We design LLM-enhanced POI embeddings, which integrate POI category distributions with natural language descriptions of each POI type, utilizing the representational power of LLMs to effectively capture the functional attributes of locations.

- We develop prediction retrieval module that employs a KD-tree to convert output coordinates into the top-k most likely predicted locations, ensuring structured outputs.

- Experiments conducted on multiple datasets validate NextLocLLM's competitive performance in both fully supervised and zero-shot scenarios.

## 2 RELATED WORK

Early methods of next location prediction largely relied on feature engineering and domain expertise. They employed features tied to urban mobility patterns, using statistical techniques and time series analysis to model and predict movement. (Noulas et al., 2012) proposed features capturing transitions between venues and spatiotemporal check-in patterns, which were then applied in linear regression. (Chen et al., 2014) introduced Next Location Predictor with Markov Modeling, incorporating individual and collective movement behaviors. Similarly, (Ying et al., 2014) employs factors in geographic, temporal, and semantic signals to predict location likelihood. These methods rely extensively on feature engineering, which often demands substantial effort and yields restricted performance.

In deep learning era, researchers adopted models like RNNs and Transformers for next location prediction, which excel in capturing temporal dependencies. STRNN (Liu et al., 2016) modeled local spatial and temporal contexts using transition matrices based on time intervals and geographic distances. (Fan et al., 2018) combined CNNs and bidirectional LSTMs for prediction by integrating contextual information. GETNext (Yang et al., 2022) introduced a Graph Enhanced Transformer that utilized global trajectory flow map. (Hong et al., 2023) developed an MHSA-based model that leverages raw location sequences, temporal data, and land use functions. (Yao et al., 2023) combined geographic embeddings, multi-layer attention, and Bi-LSTM, and integrated geographic information. While these models achieve excellent performance on specific datasets, their generalization capability remains a challenge, as they often underperform on datasets from unseen cities or scenarios.

In recent years, with the swift advancement of large language models (LLMs) (Touvron et al., 2023b;a; Achiam et al., 2023), researchers have begun to extend their researches in next location prediction with LLMs. Current studies mainly rely on specifically designed prompt engineering. (Wang et al., 2023) introduced the concepts of historical and contextual stays to capture long- and short-term dependencies in human mobility, incorporating time-aware predictions with temporal data. (Wang et al., 2024) leveraged the semantic perception capabilities of LLMs to extract personalized activity patterns from historical data. (**?**) developed a context-based chain-of-thought prompting technique, aligning LLMs with context-aware mobility behaviors through contextual learning from small samples. Despite some success, these methods exhibit notable limitations. First, these models employ discrete location IDs that neglect geographic distances and spatial relationships, thereby compromising prediction accuracy (Liu et al., 2016). Second, using location IDs hinders model transferability across cities, since identical IDs often correspond to entirely different locations in different cities, obstructing cross-city generalization (Jiang et al., 2021). Furthermore, these purely prompt-based models neglect the distribution of point of interest (POI) categories, which are crucial for capturing the functional characteristics of locations (Wang et al., 2023; Liang et al., 2024).

To address these limitations, we propose NextLocLLM, the first known model to integrate LLM directly into next location prediction structure. Unlike traditional approaches that rely on discrete location IDs, NextLocLLM uses normalized spatial coordinates, which better capture geographical distances between locations and enhance transferability and generalization across diverse urban settings. Additionally, NextLocLLM incorporates LLM-enhanced POI embeddings to more effectively model locations' functional attributes, resulting in a more comprehensive predictive framework.

## 3 PROBLEM FORMULATION

Let $L = \{loc_1, \cdots, loc_p\}$ be the set of locations, and $SD = \{SD_1, \cdots, SD_q\}$ be the set of possible stay durations. $TI = \{(d, t)\}$ is the set of temporal information, where $d$ represents day-of-week ($0 \leq d \leq 7$) and t is time-of-day ($0 \leq t \leq 23$ in hours). Each location $loc \in L$ is represented as a tuple $(id, x, y, poi)$, where $id$ is a discrete identifier, $x$ and $y$ are the spatial coordinates of location's centroid, and $poi$ represents the location's POI attributes (see Definition 2.2).

*Definition 2.1 (**Visiting Record**).* A record is defined as a tuple $s = (loc, (d, t), dur) \in L \times TI \times SD$, indicating that a user visited location $loc$ on day $d$ at hour $t$, and stayed for a duration of $dur$.

*Definition 2.2 (**POI Attributes**).* The POI attributes for each location $loc$ are represented as $attr = (intr, freq)$, where $intr = (i_1, \cdots, i_r)$ denotes the natural language descriptions of $r$ POI categories (see Appendix C), and $freq = (f_1, \cdots, f_r)$ represents the frequency of each POI category within that location. Here $r$ is the number of POI categories.

*Definition 2.3 (**Historical and Current Trajectories**).* A user's mobility trajectory can be divided into historical and current trajectories. The historical trajectory $S_h = (s_{t_1-M+1}, \cdots, s_{t_1})$ contains $M$ records, reflecting the user's long-term movement patterns. The current trajectory $S_c = (s_{t-N+1}, \cdots, s_t)$ contains $N$ records, representing the user's current moving intentions. Here $t_1 < t - N + 1$, and typically $N < M$. Since both types of trajectories share the same structure, without ambiguity, we will use $l_{seq}$ to refer to either $N$ or $M$ in subsequent sections.

*Definition 2.4 (**Next Location Prediction**).* Given a user's historical trajectory $S_h$ and current trajectory $S_c$, our target is to predict the ID of next location $loc_{t+1}$ that the user is most likely to visit.

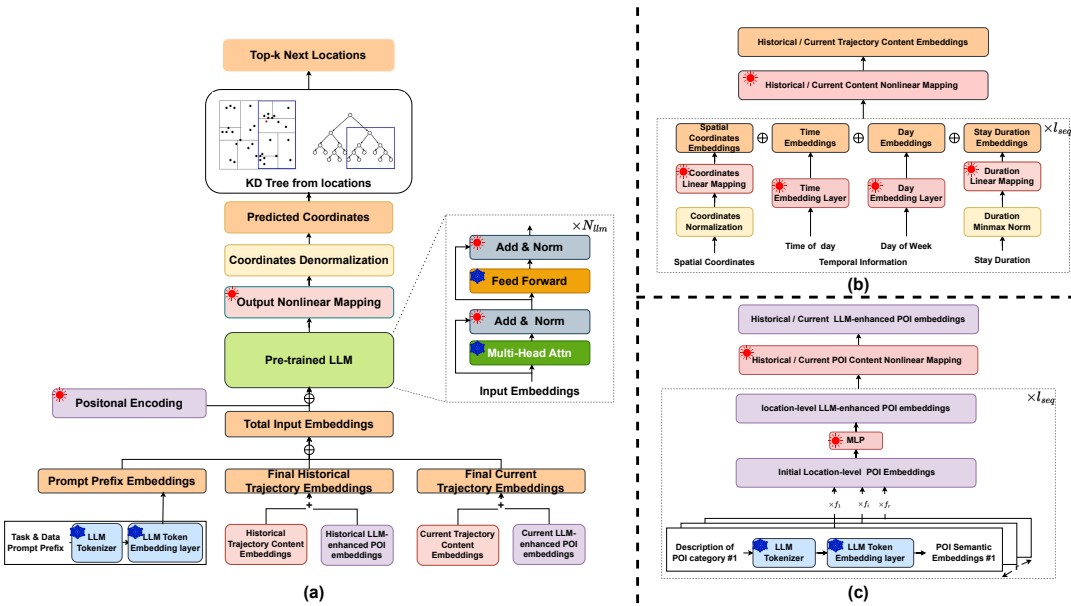

Figure 2: Structure of NextlocLLM. (a) shows its overall structure, (b) represents multi-dimensional trajectory content embeddings, (c) is the LLM-enhanced POI embeddings.

# 4 METHOD

Figure 2 illustrates an overview of our proposed NextLocLLM, which consists of four main components: (1) multi-dimensional trajectory content embeddings, (2) LLM-enhanced POI embedding, (3) an LLM backbone, and (4) a prediction retrieval module. First, we encode mobility trajectories using features including spatial coordinates, temporal information, and stay duration, to generate trajectory content embeddings. In addition, leveraging LLMs' comprehension abilities, we generate semantic embeddings for each POI category based on their natural language descriptions. These semantic embeddings are then aggregated into a weighted sum, where the frequency of each POI category serves as weights, thus effectively capturing functional attributes of locations and producing LLM-enhanced POI embeddings. Moreover, we design prompt prefix to enhance LLM's understanding of both the prediction task and the data structure. The prompt prefix, combined with trajectory content embeddings and LLM-enhanced POI embeddings, are then passed into the LLM backbone. Finally, the predicted coordinates from the LLM are processed through the prediction retrieval module, which employs a KD-tree to retrieve the $k$ nearest candidate locations, ensuring structured prediction results.

## 4.1 MULTI-DIMENSIONAL TRAJECTORY CONTENT EMBEDDINGS

For each mobility trajectory, we extract features for each record from both the historical trajectory $S_h$ and the current trajectory $S_c$. These features include spatial coordinates $((x, y)_h, (x, y)_c)$, time-of-day $(T_h, T_c)$, day-of-week $(D_h, D_c)$ and stay duration $(Dur_h, Dur_c)$. Each sequence is processed using embedding or linear mapping functions, with the same function applied to both historical and current sequences of the same feature type. Specifically, we develop embeddings for time-of-day and day-of-week, denoted as $f_t$ and $f_d$. Additionally, we utilize linear mappings for spatial coordinates and stay duration, represented by $f_{xy}$ and $f_{dur}$, to transform raw data into compact embeddings.

**Temporal and Stay Duration Embeddings**. To capture temporal characteristics of mobility trajectories, we employ independent embeddings for two critical temporal information: time-of-day ($T$) and day-of-week ($D$). These temporal aspects are crucial, as user mobility patterns are often strongly correlated with specific times and days. The temporal features are transformed into embedding vectors through look-up operations from their respective embedding tables, where time-of-day embeddings and day-of-week embeddings are generated by $E_T = f_t(T) \in \mathbb{R}^{l_{seq} \times d_t}$ and $E_D = f_d(D) \in \mathbb{R}^{l_{seq} \times d_d}$, respectively. In addition, stay duration is another critical temporal dimension for understanding user behavior. Typically, longer stay durations indicate more complex activities, such as working or dining, while shorter durations are associated with quick actions or transitory activities. The stay durations are processed with min-max normalization, and then transformed into embeddings through linear mapping function $E_{Dur} = f_{dur}(Dur) \in \mathbb{R}^{l_{seq} \times d_{dur}}$.

**Spatial Coordinates Embeddings**. Existing methods typically use numerical identifiers as location IDs to represent and encode location information. However, these discrete IDs neither capture the geographical relationships between locations nor enable effective transferability across different cities. Thus, we propose using spatial coordinates instead of discrete IDs to represent locations' geographic information. Specifically, we utilize Web Mercator coordinates, as they can reflect the spatial relationships between locations. At city level, the gap between geodesic distance and Euclidean distance under Web Mercator projection is minimal, making it suitable for practical applications (Battersby et al., 2014; Peterson, 2014) (see Appendix D, E). While Web Mercator coordinates effectively capture spatial relationships within cities, the range in coordinates vary significantly between cities, which presents generalization challenges in zero-shot scenarios. To mitigate this, we apply normalization by scaling them to a standard normal distribution $N(0, 1)$. The mean and variance for it are based on spatial coordinates from all mobility trajectory records in the target city, rather than the geographical data, as high population density areas are not always located at the geometric center of a city (see Fig. 4). Finally, the normalized coordinates $(x', y')$ are transformed into spatial coordinates embeddings with a linear function: $E_{XY} = f_{xy}(x', y') \in \mathbb{R}^{l_{seq} \times d_{xy}}$.

**Trajectory Content Embeddings**. So far, we have obtained spatial coordinates embeddings $E_{XY}$, time embeddings $E_T$, day embeddings $E_D$, and stay duration embeddings $E_{Dur}$. To integrate these features, we first concatenate these embeddings for each trajectory to generate the combined content embeddings:

$$E_{all_h} = E_{XY_h} || E_{T_h} || E_{D_h} || E_{Dur_h} \in \mathbb{R}^{M \times (d_{xy} + d_t + d_d + d_{dur})} \tag{1}$$

$$E_{all_c} = E_{XY_c} || E_{T_c} || E_{D_c} || E_{Dur_c} \in \mathbb{R}^{N \times (d_{xy} + d_t + d_d + d_{dur})} \tag{2}$$

After we obtain the combined content embeddings, we apply nonlinear functions $f_h$ and $f_c$, parameterized by multi-layer perceptrons (MLP), to convert these embeddings into LLM's embedding dimension $d_{llm}$ for the historical trajectory and the current trajectory respectively:

$$E_{con_h} = f_h(E_{all_h}) \in \mathbb{R}^{M \times d_{llm}}, E_{con_c} = f_c(E_{all_c}) \in \mathbb{R}^{N \times d_{llm}} \tag{3}$$

## 4.2 LLM-ENHANCED POI EMBEDDING

In next location prediction, understanding the functional attributes of a location is crucial. The functionality of a location is determined by the distribution of various points-of-interest (POIs) within that area. Prior research (Hong et al., 2023; Yang et al., 2022) has demonstrated that incorporating POI information offers deeper insights into location characteristics, helping to capture user preferences and behaviors.

To address this issue, we propose LLM-enhanced POI embeddings, leveraging LLMs to better represent the functional attributes of locations. Specifically, for the natural language descriptions

of POI categories $intr = (i_1, \cdots, i_r)$, we generate corresponding POI semantic embeddings $E_I = (E_1, \cdots, E_r) \in \mathbb{R}^{r \times l \times d_{llm}}$ through the token embedding layer of LLM, where $l$ is a predefined description length. The POI distribution $freq = (f_1, \cdots, f_r)$ at each location is then used as weights to perform a weighted summation of the corresponding POI semantic embeddings, resulting in the initial location-level POI embeddings:

$$E_{POI_{loc,init}} = \sum_{j=1}^{r} E_j * f_j \in \mathbb{R}^{lt \times d_{llm}} \tag{4}$$

To further integrate such POI semantic information, we apply a nonlinear transformation with MLP to obtain the final location-level LLM-enhanced POI embeddings:

$$E_{POI_{loc}} = \text{MLP}(E_{POI_{loc,init}}) \in \mathbb{R}^{d_{llm}} \tag{5}$$

For mobility trajectories, we concatenate the location-level POI embeddings for each record, producing the initial trajectory-level LLM-enhanced POI embeddings $E_{L-POI} \in \mathbb{R}^{l_{seq} \times d_{llm}}$. Additionally, such trajectory-level LLM-enhanced POI embeddings for historical and current trajectory are separately passed through $f_{h_{poi}}$ and $f_{c_{poi}}$ with MLP, to accommodate the distinct temporal semantics of historical and current trajectories as the LLM-enhanced POI embeddings for each type of trajectory:

$$E_{L-POI_h} = f_{h_{poi}}(E_{L-POI}) \in \mathbb{R}^{M \times d_{llm}}, E_{L-POI_c} = f_{c_{poi}}(E_{L-POI}) \in \mathbb{R}^{N \times d_{llm}} \tag{6}$$

## 4.3 LARGE LANGUAGE MODEL BACKBONE

**Total Input Embeddings** Given the the content embeddings $E_{con_h}$ and $E_{con_c}$ representing historical and current trajectories respectively, and the corresponding LLM-enhanced POI embeddings $E_{L-POI_h}$ and $E_{L-POI_c}$ that capture the functional attributes of various locations, we combine these two embeddings to form the final embeddings for both the historical and current trajectories:

$$E_{his} = E_{con_h} + E_{L-POI_h} \in \mathbb{R}^{M \times d_{llm}}, E_{cur} = E_{con_c} + E_{L-POI_c} \in \mathbb{R}^{N \times d_{llm}} \tag{7}$$

To further improve the model's comprehension of the input data and prediction task, we craft a task- and data-specific prompt prefix that defines the task, describes the data, and explains how to utilize historical and current trajectories (see Appendix H). This prompt prefix is processed through LLM's token embedding layer to generate prompt prefix embeddings. Finally, we concatenate prompt prefix embeddings with the two final embeddings to form the total input embeddings for the LLM:

$$E_{total} = E_{instruct}||E_{his}||E_{cur} \tag{8}$$

**Partially Frozen Large Model**. To fully leverage the extensive knowledge embedded within LLMs while preserving their powerful reasoning capabilities, we adopt the strategy in (Zhou et al., 2023) by freezing the self-attention and feedforward layers of the LLM. These core components retain the majority of the knowledge acquired during pre-training, and freezing them ensures that this knowledge remains intact, preventing unintended modifications during task-specific fine-tuning. We inject the task-specific knowledge for the next location prediction task by fine-tuning a small subset of parameters contained in positional encoding layers and layer normalization (Zhou et al., 2023; Liu et al., 2024). This allows LLM to quickly adapt to our task with minimal costs and resources. Specifically, given the total input embedding $E_{total}$, the LLM produces the output representation $E_o = \text{LLM}(E_{total})$. We extract the last vector $\mathbf{v}_o \in \mathbb{R}^{d_{llm}}$ from $E_o$ at the end of the sequence, and apply an nonlinear function $f_o$ to generate the predicted spatial coordinates $xy'_o = f_o(o)$. This result is then converted back to te original scale to obtain the denormalized coordinates $xy_o$. During training, the objective is to minimize the Euclidean distance between $xy_o$ and ground truth $\hat{xy}$.

## 4.4 PREDICTION RETRIEVAL MODULE

During inference, after obtaining the predicted coordinates $xy_o$, we employ a prediction retrieval module to determine the most likely top-k locations, to produce structured next location prediction. Specifically, we construct a KD-tree based on spatial coordinates of all candidate locations' centers. By using $xy_o$ as the query point to this KD-tree, we retrieve the top-k locations closest to the predicted coordinates. These locations are then returned as the final top-k location predictions of NextLocLLM.

Table 1: Fully Supervised Next Location Prediction Result

| Method | Xi'an | | | Chengdu | | | Japan | | |
|---|---|---|---|---|---|---|---|---|---|
| | Hit@1 | Hit@5 | Hit@10 | Hit@1 | Hit@5 | Hit@10 | Hit@1 | Hit@5 | Hit@10 |
| STRNN | 11.01% | 19.15% | 25.61% | 22.40% | 32.09% | 37.45% | 8.616% | 15.45% | 24.46% |
| LSTM | 9.753% | 31.17% | 45.34% | 17.48% | 46.93% | 63.40% | 2.817% | 9.993% | 15.67% |
| FPMC | 20.97% | 39.95% | 47.58% | 20.94% | 49.91% | 62.46% | 2.973% | 7.859% | 13.51% |
| GRU | 9.590% | 30.92% | 45.17% | 16.41% | 47.80% | 63.06% | 3.831% | 10.69% | 15.56% |
| C-MHSA | 50.32% | 92.43% | 95.38% | **76.54%** | 97.44% | 99.37% | **20.17%** | 30.23% | 37.68% |
| DeepMove | 41.19% | 83.02% | 90.85% | 57.99% | 94.71% | 98.38% | 11.71% | 22.23% | 36.35% |
| GETNext | 48.63% | 85.67% | 93.25% | 72.57% | 98.29% | 99.56% | 19.17% | 28.62% | 33.79% |
| LLMMob(wt) | 33.52% | 77.86% | 78.00% | 45.27% | 81.65% | 84.37% | 17.63% | 28.55% | 37.26% |
| LLMMob(wt,s) | 20.81% | 62.08% | 62.23% | 26.63% | 59.97% | 62.26% | 12.26% | 21.87% | 31.19% |
| LLMMob(wot) | 31.27% | 72.49% | 73.31% | 41.13% | 80.06% | 82.21% | 17.29% | 27.26% | 27.40% |
| LLMMob(wot,s) | 16.77% | 58.93% | 59.12% | 23.37% | 57.71% | 59.36% | 12.57% | 21.62% | 20.87% |
| ZS-NL | 20.92% | 53.29% | 66.99% | 31.06% | 62.25% | 64.47% | 13.07% | 22.31% | 26.15% |
| ZS-NL(s) | 20.27% | 52.22% | 64.97% | 26.78% | 49.57% | 54.67% | 11.32% | 19.15% | 23.57% |
| NextlocLLM | **58.14%** | **97.14%** | **99.36%** | 64.33% | **98.48%** | **99.63%** | 19.36% | **31.82%** | **46.06%** |

## 5 EXPERIMENT

In this section, we evaluate the performance of NextLocLLM in comparison with existing methods under both fully-supervised and zero-shot next location prediction scenarios. In fully-supervised scenarios, each model is tested on data from the same cities as the training set, whereas in zero-shot scenarios, each model is tested on data from cities that are not part of the training set, requiring the model to generalize based on knowledge learned from other cities. The code for NextLocLLM is available at: `https://anonymous.4open.science/r/NexelocLLM-1CF8/`

### 5.1 EXPERIMENTAL SETUP

#### 5.1.1 BASELINE MODELS

We selected several classical baseline models as well as recently proposed methods for comparison. These models include those not designed for zero-shot scenarios (LSTM, FPMC, GRU, STRNN, C-MHSA, DeepMove, and GETNext) and models that support zero-shot next location prediction (LLMMob and ZS-NL). Detailed descriptions of these baseline models are provided in Appendix. G.

#### 5.1.2 DATASETS

We used four user mobility trajectory datasets in our experiments, including three open-source datasets—Xi'an, Chengdu (Zhu et al., 2023), and Japan (Yabe et al., 2024b)—and one private dataset from Singapore. These datasets cover diverse geographical regions and user behavior patterns, ensuring our results to be broadly applicable. Detailed descriptions can be found in Appendix I.

#### 5.1.3 EVALUATION METRICS

We employed Hit@1, Hit@5, and Hit@10 as the evaluation metrics for model performance. These metrics measure the proportion of cases in which the model correctly predicts the target location within the top-k predictions. Specifically, they indicate the accuracy of models when the correct location appears in the top 1, top 5, or top 10 predictions.

#### 5.1.4 EXPERIMENT CONFIGURATION

For each dataset, we split the data for training, validation, and testing with a 70%/10%/20% ratio. For zero-shot tasks, we only used the final 20% test set for evaluation. For LLM backbone in NextLocLLM, GPT-2 is employed unless otherwise specified. We also utilized other LLMs (Llama2-7B and Llama3-8B) as the backbone for NextLocLLM and found that their prediction performances are relatively consistent compared to using GPT-2. Detailed results can be found in Appendix N.

## 5.2 Fully Supervised Next Location Prediction Performance Comparison

Table 1 presents the performance of different models on next location prediction tasks in fully-supervised scenarios. The table is divided into three sections by two horizontal lines. Models above the first horizontal line are designed for fully-supervised scenarios, while models above the second horizontal line can be used in both fully-supervised and zero-shot settings.

In the first group, models like STRNN, LSTM, FPMC, and GRU struggle to effectively capture temporal dependencies, leading to relatively poor performance. In contrast, models that utilize attention mechanisms, such as C-MHSA, DeepMove, and GETNext, demonstrate significantly better predictive performance. The attention mechanism allows these models to better capture complex dependencies across different time points, thereby improving prediction accuracy.

In the second group, LLMMob and ZS-NL interact with LLMs purely through prompts. Compared with ZS-NL, LLMMob provides more detailed task instructions to guide LLMs, thus outperforming ZS-NL. LLMMob (wt) and LLMMob (wot) denote whether temporal information is considered. The results show that models including temporal information (wt) outperform those without (wot), indicating that time is a crucial feature in next location prediction and significantly enhances model accuracy. Additionally, the suffix "s" indicates a strict requirement for the LLM to output exactly 10 location IDs. If this requirement is not met, even if the correct location is predicted, it does not count toward the accuracy. Conversely, configurations without the "s" suffix relax this restriction. This experimental setup was based on our observation that, even when the prompt clearly specifies the output format and the required number of location IDs, these models sometimes generate outputs with incorrect formats or numbers of IDs (see Sec Fig. 3 for details). Results show that enforcing strict output requirements leads to a significant performance drop for both LLMMob and ZS-NL.

Furthermore, we observe that all models perform better on Xi'an and Chengdu datasets compared to Japan dataset. We attribute this to the fact that Xi'an and Chengdu datasets have more users, a longer time span, and shorter average sampling intervals. These factors provide richer mobility information for the models, enabling them to better learn travel patterns and improve performance.

Our proposed NextLocLLM consistently outperforms all baseline models across these datasets. NextlocLLM leverages spatial coordinates to represent locations, which allows it to better model spatial relationships between locations. Additionally, it uses LLM-enhanced POI embeddings to fully exploit the large model's natural language understanding capabilities, capturing the functional attributes of locations more effectively. On Xi'an dataset, NextLocLLM achieved the highest scores. Even on Chengdu and Japan datasets, where Hit@1 was slightly lower, NextLocLLM still outperformed other models in Hit@5 and Hit@10, showcasing strong predicting capabilities.

Additionally, we conducted experiments on a private dataset from Singapore. This dataset is relatively sparse and of lower quality, resulting in poor performance for all models. However, even in this sparse and low-quality data environment, NextLocLLM maintained competitive performance, outperforming other models. Detailed results and analysis for the Singapore dataset can be found in Appendix L

## 5.3 Zero-shot Next Location Prediction Performance Comparison

Table 2 presents the performance of different models in zero-shot scenarios. We compared LLMMob, ZS-NL, and our proposed NextLocLLM. For NextLocLLM, we trained the model on Singapore, Chengdu, and Japan datasets, and then directly tested it on Xi'an dataset. In addition to accuracy metrics, we also calculated the geographical distance between predicted coordinates and actual location centers to further evaluate NextLocLLM's precision. For reference, the average geographical distance in the fully-supervised scenario for NextLocLLM on the Xi'an dataset is 176.9 meters.

The results show that well-trained NextLocLLM models demonstrate strong cross-city generalization. Specifically, NextLocLLM (Chengdu -> Xi'an) and NextLocLLM (Japan -> Xi'an) outperformed other zero-shot models, demonstrating robust cross-city adaptability. In contrast, NextLocLLM (Singapore -> Xi'an) performed the weakest, likely due to the lower quality of the Singapore dataset, which hindered proper training and limited the model's ability to generalize in the zero-shot scenario.

The geographical distance results further support this analysis. For NextLocLLM (Chengdu -> Xi'an), the average prediction error was 449.79 meters, which is close to the 176.9 meters observed in fully-supervised scenario and smaller than the 500-meter grid size for locations, indicating high

Table 2: Zero-shot Next Location Prediction Result on Xi'an Dataset

|  | Hit@1 | Hit@5 | Hit@10 | Distance(s) |
|---|---|---|---|---|
| LLMMob(wt) | 33.52% | 77.86% | 78.00% | - |
| LLMMob(wt,s) | 20.81% | 62.08% | 62.23% | - |
| LLMMob(wot) | 31.27% | 72.49% | 73.31% | - |
| LLMMob(wot,s) | 16.77% | 58.93% | 59.12% | - |
| ZS-NL | 20.92% | 53.29% | 66.99% | - |
| ZS-NL(s) | 20.27% | 52.22% | 64.97% | - |
| NextlocLLM(Singapore->Xi'an) | 3.85% | 9.55% | 19.86% | 4521.74 |
| NextlocLLM(Chengdu->Xi'an) | 37.02% | 82.26% | 92.41% | 449.79 |
| NextlocLLM(Japan->Xi'an) | 27.38% | 75.87% | 86.84% | 765.73 |

Table 3: Ablation Study for NextLocLLM Modules

| Prompt Prefix | LORA | LLM-enhanced POI | Hit@1 | Hit@5 | Hit@10 |
|---|---|---|---|---|---|
| ✗ | ✗ | ✗ | 25.81% | 83.20% | 97.54% |
| ✗ | ✗ | ✓ | 45.79% | 94.76% | 99.24% |
| ✗ | ✓ | ✗ | 16.18% | 65.29% | 89.16% |
| ✗ | ✓ | ✓ | 38.11% | 81.78% | 86.87% |
| ✓ | ✗ | ✗ | 39.57% | 90.81% | 98.75% |
| ✓ | ✓ | ✗ | 27.57% | 78.46% | 98.75% |
| ✓ | ✓ | ✓ | 28.34% | 77.16% | 89.27% |
| ✓ | ✗ | ✓ | **58.14%** | **97.14%** | **99.36%** |

geographical accuracy. In contrast, the average prediction error for NextLocLLM (Singapore -> Xi'an) is 4521.74 meters, far exceeding that of other models. In summary, well-trained NextLocLLM significantly outperform other baseline models in zero-shot scenarios.

We also compared the zero-shot performance of these models on other datasets, with similar conclusions as those observed on the Xi'an dataset. Detailed results can be found in Appendix M.

## 5.4 ABLATION STUDY

To validate the effectiveness of different components in NextLocLLM, we conducted a series of ablation experiments in this section.

### 5.4.1 ABLATION STUDY FOR NEXTLOCLLM KEY COMPONENTS

We first evaluated the impact of adding prompt prefix, using LLM-enhanced POI embeddings, and freezing most of LLM's parameters versus using fine-tuning methods like LoRA. Specifically, not using LLM-enhanced POI embeddings means treating the POI distribution of each location as a simple numerical vector and applying a linear mapping directly to generate the POI embeddings $E_{POI} \in \mathbb{R}^{d_{poi}}$. In this case, equations (1) and (2) will change as follows:

$$E_{all_h} = E_{XY_h}||E_{T_h}||E_{D_h}||E_{Dur_h}||E_{POI_h} \in \mathbb{R}^{M \times (d_{xy}+d_t+d_d+d_{dur}+d_{poi})} \tag{9}$$

$$E_{all_c} = E_{XY_c}||E_{T_c}||E_{D_c}||E_{Dur_c}||E_{POI_c} \in \mathbb{R}^{N \times (d_{xy}+d_t+d_d+d_{dur}+d_{poi})} \tag{10}$$

Moreover, due to the absence of LLM-enhanced POI embeddings, the final embeddings for historical and current trajectories, $E_{con_h}$ and $E_{con_c}$, are simply just their content embeddings $E_{con_h}$ and $E_{con_c}$.

Table 4: Ablation Study for NextLocLLM Input Components

| Day & Time | Duration | POI | Hit@1 | Hit@5 | Hit@10 |
|---|---|---|---|---|---|
| ✓ | ✓ | ✗ | 52.78% | 95.29% | 98.55% |
| ✓ | ✗ | ✓ | 47.65% | 94.94% | 98.23% |
| ✗ | ✓ | ✓ | 48.28% | 96.00% | 98.25% |
| ✓ | ✓ | ✓ | **58.14%** | **97.14%** | **99.36%** |

Table 5: Ablation Study for Historical and Current Trajectory

| Historical Trajectory | Current Trajectory | Hit@1 | Hit@5 | Hit@10 |
|:---:|:---:|:---:|:---:|:---:|
| ✓ | ✗ | 18.75% | 60.62% | 82.10% |
| ✗ | ✓ | 40.23% | 91.94% | 98.66% |
| ✓ | ✓ | **58.14%** | **97.14%** | **99.36%** |

As shown in Table 3, models using prompt prefix consistently outperform those without, showing the positive effect of prompt prefix on enhancing LLMs' understanding of data and task. Additionally, models that incorporates LLM-enhanced POI embedding outperforms the ones without it, highlighting the importance of using LLM for encoding natural language descriptions of POI categories to better model location functionality. Finally, the configuration where most large language model parameters are frozen performs better than the fully fine-tuning approach, indicating that freezing core modules of LLMs and finetuning the rest helps preserve its pre-trained knowledge while adapting to the specific task with minimal parameter adjustments. Overall, the best configuration for NextLocLLM combines prompt prefix, LLM-enhanced POI embedding, and freezing the major LLM parameters.

### 5.4.2 ABLATION STUDY FOR NEXTLOCLLM INPUT COMPONENTS

We further evaluated the effectiveness of temporal information, stay duration, and POI attributes. Results are presented in Table 4, which clearly demonstrate that introducing time, stay duration, and POI functionality information is crucial for the predictive performance of NextLocLLM. Removing any of these elements results in a performance drop, with temporal information and stay duration showing particularly significant effects on user behavior prediction. The complete model, which integrates these various features, provides a more comprehensive and accurate reference for predictions.

### 5.4.3 ABLATION STUDY FOR HISTORICAL AND CURRENT TRAJECTORY

We also evaluate the impact of historical and current trajectory on the performance of NextLocLLM for next location prediction. To this end, we conducted experiments where either the historical trajectory or the current trajectory was used as the sole input. The results are presented in Table 5. The findings indicate that combination of both trajectories significantly improves prediction accuracy. Using only the historical trajectory or only the current trajectory leads to a drop in performance. The historical trajectory reflects the user's long-term behavior patterns, while the current trajectory captures short-term behavioral intentions. Combining both allows NextlocLLM to more comprehensively model user behavior, thereby improving the accuracy of next location predictions.

## 6 CONCLUSION

In this paper, we present NextLocLLM, the first known method to integrate large language models (LLMs) into the structure of next location prediction models. By innovatively encoding normalized spatial coordinates, NextLocLLM effectively captures geographic relationships between locations and its ability to transfer across different cities is significantly enhanced. Additionally, NextLocLLM incorporates LLM-enhanced POI embeddings, leveraging LLMs to encode natural language descriptions of point-of-interest (POI) categories, enabling itself to better understand the functional characteristics of locations. To reduce training costs while preserving the pre-trained knowledge, we freeze most of the LLM's parameters and fine-tune only a few key layers. This strategy ensures that LLMs can quickly adapt to next location prediction task. The prediction retrieval module guarantees structured top-k location predictions. Experimental results demonstrate that NextLocLLM consistently outperforms baseline models in both fully-supervised and zero-shot prediction scenarios, showing strong power in next location prediction task. Ablation studies confirm the contribution of each key module to the overall performance. However, we observe that the current model's geographical distance error remains a challenge, typically exceeding 200 meters, which limits its effectiveness in fine-grained prediction tasks, such as those with a 50-meter grid size. Future work will explore incorporating user profile information and other essential elements to further improve accuracy. Overall, NextLocLLM offers an innovative and effective solution for next location prediction, with significant potential for real-world applications across various domains.

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

## A  PROMPT FOR ZS-NL AND LLM-MOB

In this section, we present the prompt formats used by the baseline models ZS-NL and LLM-Mob. We preserve the original prompt content from the source papers in full. We highlighted in red the parts of each model's prompt that specify the required number of location IDs to be output. Additionally, to clearly differentiate between LLM-Mob(wt) and LLM-Mob(wot), the extra components in LLM-Mob(wt) are marked in blue.

## B  EXAMPLE FOR WRONGLY STRUCTURED OUTPUT BY PURELY PROMPTED-BASED NEXT LOCATION PREDICTION MODEL

Fig 3 shows examples of LLM-Mob on the Xi'an dataset. We can observe that for line index 1, the model only produced three outputs, even though the prompt clearly instructed the LLM to provide 10 possible location IDs. This discrepancy highlights a common issue with prompt-based models: they sometimes fail to follow explicit prompt prefixes regarding output structure. In this case, despite the requirement for 10 locations, the model returned fewer results, which could affect the overall performance. This inconsistency raises concerns about the reliability of prompt-based methods when dealing with structured output requirements, particularly in tasks that demand a specific number of predictions.

| | user_id | ground_truth | prediction | reason |
|---|---|---|---|---|
| 0 | 662041 | 705 | [706, 705, 707, 708, 709, 710, 711, 712, 713, 714] | The user has a strong pattern of staying at place 706 during the evening hours on Tuesday, with multiple visits around 6 PM to 8 PM. Given that the target stay is at 10 PM on the same day, it is likely that the user will continue to stay at a familiar location. Place 705 is also a recent location visited at 8 PM, making it a probable choice as well. Other places (707-714) are included as potential options based on general evening activity patterns. |
| 1 | 662041 | 731 | [705, 731, 706] | The user has a clear pattern of visiting places 705 and 731 during the evening hours on Tuesday, particularly around 8 PM and 10 PM. Given that the target stay is at 12 AM on Tuesday, it is likely that the user would continue their activities at one of these places. Place 706 was visited earlier in the evening but less frequently, making it a less likely option compared to 705 and 731. |
| 2 | 662041 | 759 | [731, 732, 733, 734, 735, 736, 737, 738, 739, 740] | The user has a strong pattern of staying at place ID 731 during late-night hours on Tuesday and early Wednesday. Given the target stay is at 3:00 AM on Wednesday, it is highly probable that the user will continue to stay at the same location (731) or nearby places (732-740) that are likely to be open or frequented during this time. |
| 3 | 685968 | 541 | [576, 101, 102, 103, 104, 105, 106, 107, 108, 109] | The user has a strong pattern of staying at place ID 576 during the evening on Mondays, as evidenced by multiple recent visits around 6:00 PM and 7:00 PM. Given the target stay is also on a Monday at 7:00 PM, it is highly probable that the user will return to place ID 576. Other places are ranked based on their proximity or similarity to the user's recent activities, but 576 is the most likely due to the frequency of visits. |

Figure 3: Part of LLM-MOB's output on Xi-an Dataset.

Table 6: POI categories' description (For Xian and Chengdu)

| POI category | Description |
| --- | --- |
| Entertainment | Entertainment: This category combines scenic spots with sports and recreation services for leisure activities. |
| Commercial | Commercial: It includes businesses, financial services, automotive, shopping, and dining services. |
| Education | Education: This category covers institutions which involved in science, education, and cultural services. |
| Public Service | Public Service: including government, daily services, healthcare, transport, and public infrastructure. |
| Residential | Residential: This category comprises accommodation services and mixed-use commercial and residential areas. |

Table 7: POI categories' description (For Sinapore)

| POI category | Description |
| --- | --- |
| Leisure and Entertainment | Leisure and Entertainment: This category encompasses venues for arts, entertainment, events, and nightlife activities, serving as hubs for cultural, social, and recreational engagements. |
| Shopping and Services | Shopping and Services: It includes retail outlets and professional service providers, catering to the diverse purchasing and service needs of consumers. |
| Dining and Health | Education: This category covers eating establishments with health and medical services, offering places for dining along with health care facilities. |
| Travel and Accommodation | Travel and Accommodation: including all travel-related infrastructure and lodging options, including transportation hubs, universities, and residential areas, facilitating mobility and accommodation. |
| Outdoor and Recreational Activities | Outdoor and Recreational Activities: This category comprises outdoor spaces and landmarks, providing areas for recreation and appreciation of natural and cultural heritage. |

## C  POI CATEGORIES' DESCRIPTION

In this section, we provide descriptions of the different POI categories across various datasets. The raw POI distribution for the Xi'an and Chengdu datasets are sourced from (Center, 2017), while the Japan dataset comes with its own raw POI distribution, and the raw POI distribution data for the Singapore dataset was scraped from openstreetmap. The original POI categories were quite numerous, resulting in very few POIs for each location and making the vectors extremely sparse. Additionally, the definitions and classifications of POIs vary significantly across the datasets. To address these issues, we clustered POI categories with similar attributes, aggregated the number of POIs in each category for each location, and provided natural language descriptions for each clustered POI category.

## D  TRANSFORMATION FROM LONGITUDE & LATITUDE TO WEB MERCATOR

Given a location's geographic coordinates $(lon, lat)$, we can convert them into Web Mercator coordinates using the following formulas:

$$lon_{wm} = lon \times \frac{R}{D} \tag{11}$$

$$lat_{wm} = \ln(\tan(\frac{\pi}{4} + \frac{lat}{2})) \times \frac{R}{D} \tag{12}$$

where $R$ represents half the Earth's circumference, which is approximately $20,037,508.34$ meters. The constant $D$ represents 180 degrees, as the Earth's longitude ranges from $-180°$ to $180°$.

## E   ANALYSIS OF WEB MERCATOR PROJECTION'S LITTLE BIAS IN URBAN AREAS

The bias of the Web Mercator projection increases with latitude, but within smaller regions (such as city scales), this bias is relatively minimal, particularly for cities located in low- and mid-latitude areas (Battersby et al., 2014; Peterson, 2014; Hwang, 2013). To clarify this, we can explain through mathematical analysis.

The Web Mercator projection maps spherical coordinates (longitude and latitude) onto 2D plane coordinates, with the formulas shown in Appendix D. The Mercator projection preserves angles (conformal) but not area, meaning that regions at higher latitudes (especially near the poles) are stretched in terms of area and distance. For lower and mid-latitude regions (such as most city scales), this distortion is relatively minor and can be neglected. We can demonstrate the Web Mercator projection's bias at a city scale with the following approach.

Assume the latitude range within a city is $\Delta\phi$ and the longitude range is $\Delta lat$. Since the Web Mercator projection maintains a linear relationship for longitude, the primary bias arises from the non-linear term in latitude $lat_{wm} = \ln(\tan(\frac{\pi}{4} + \frac{lat}{2})) \times \frac{R}{D}$. If the latitude difference within the city is small (e.g., within 1°), we can analyze the error over this small range using a Taylor expansion:

$$y(lat + \Delta lat) \approx y(lat) + \frac{dy}{dlat}\Delta lat \tag{13}$$

We then calculate $\frac{dy}{dlat} = \frac{R}{cos(lat)}$. At low and mid-latitudes, this calculation shows that the error is relatively small, meaning the Web Mercator projection behaves almost linearly within this range.

## F   HEATMAP OF DIFFERENT DATA

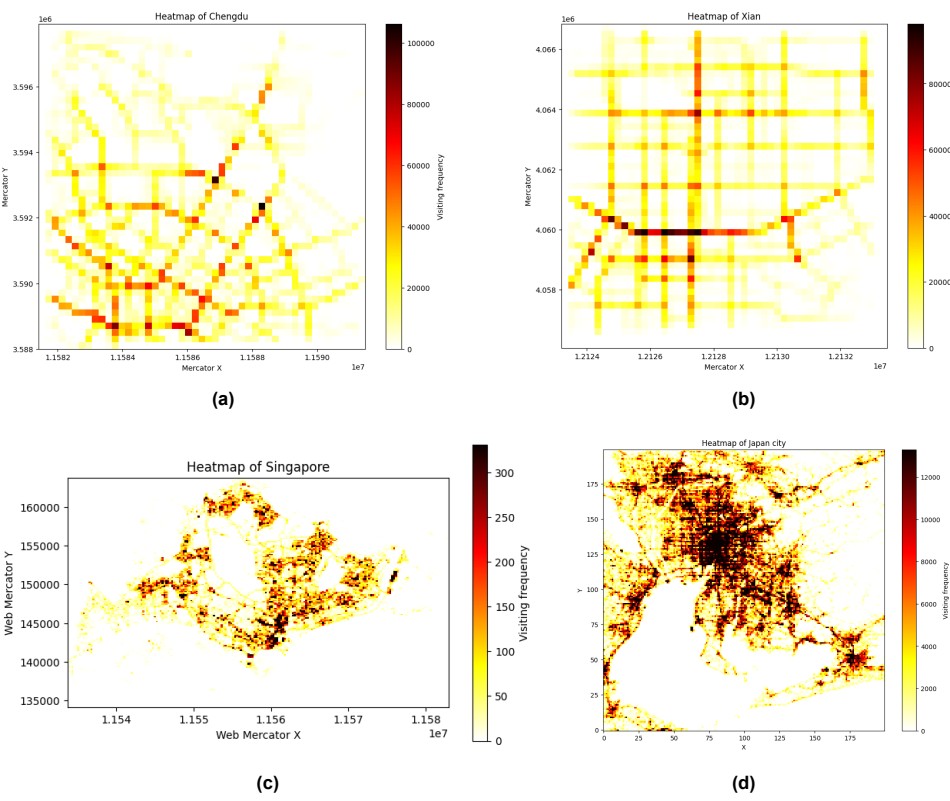

Figure 4: Heatmap for human mobility data used.

Fig. 4 presents the heatmaps for mobility data across four cities: Chengdu, Xi'an, Singapore, and a city in Japan. From these visualizations, we can observe that the centroid of each city is not always

aligned with the center of mobility data. In other words, the areas with the highest population density and movement activity often do not correspond to the geographic center of the city. This discrepancy illustrates the uneven distribution of human movement, which is influenced by factors such as data collection and formation, urban design, commercial hubs, and transportation networks.

As a result, instead of using the geographic center of the city for normalization, we rely on the mean coordinates of the mobility data. This approach better captures the actual spatial patterns and movement trends, allowing for more accurate normalization of location data. By focusing on the mobility data's mean coordinate, we can ensure that the model accurately represents the central tendency of human movement within the city, rather than relying on purely geographic or administrative boundaries.

## G  BASELINE DESCRIPTION

The details of baseline methods are briefly summarized as follows. For FPMC, LSRM, GRU, STRNN and DeepMove, we use their implementation provided by the package Libcity whereas for C-MHSA, GETNext, LLM-Mob and ZS-NL, we use the source codes released by their authors.

- FPMC (Rendle et al., 2010): a method bringing both matrix factorization (MF) and Markov chains (MC) approaches together, which is based on personalized transition graphs over underlying Markov chains.
- LSTM (Graves & Graves, 2012) A type of recurrent neural network capable of learning order dependence in sequence prediction problems.
- GRU (Chung et al., 2014) Similar to LSTMs, GRUs are a streamlined version that use gating mechanisms to control the flow of information and are effective in sequence modeling tasks.
- STRNN (Liu et al., 2016) This model focuses on introducing spatiotemporal transfer features into the hidden layer of RNN.
- Deepmove (Feng et al., 2018) This model uses the attention mechanism for the first time to combine historical trajectories with current trajectories for prediction.
- GETNext (Yang et al., 2022) An model utilizes a global trajectory flow map and a novel Graph Enhanced Transformer model to better leverage extensive collaborative signals.
- C-MHSA (Hong et al., 2023) An MHSA-based model that integrates various contextual information from raw location visit sequences.
- LLM-Mob (Wang et al., 2023) An purely prompt based model which introduced concepts of historical and contextual stays to capture the long-term and short-term dependencies in human mobility.
- ZS-NL (Beneduce et al., 2024) Another purely prompt based model designed for zero-shot next location prediction.

## H  PROMPT PREFIX PROVIDED

In this section, we outline the specific task and data prompt prefix used in NextLocLLM. The prompt prefix begins by defining the task and providing a detailed description of the dataset structure. Additionally, the Additional Description section emphasizes how to think about this task using the provided data.

## I  DATASET INTRO

We used four datasets to validate the effectiveness of NextLocLLM, and the detailed descriptions of these datasets are as follows:

- Xian & Chengdu (Zhu et al., 2023) Synthetic trajectory datasets with high fidelity. The datasets are generated by diffusion models.
- Japan (Yabe et al., 2024b) The data is about human mobility in one metropolitan area somewhere in Japan.

```
<|start_prompt|>
    Task Description: Predict the next possible location, in normalized mercator coordinates, of a resident based on their historical and
current movement trajectory.
     Data Description: This dataset includes mobility trajectory data of residents. Each record consists of historical and current
trajectories. The historical trajectory 40 records, while the current trajectory consists of 5 records, all sequentially arranged in
chronological order.
    Additional Description:
            •   Historical records effectively describe the resident's regular travel patterns and frequently visited places, while current
                trajectories better reflect the user's current location and their short-term travel intentions.
            •   Each record combines normalized mercator coordinates, day of week, time of day, duration and POI categories.
    <|end_prompt|>
```

Figure 5: Data and Task Prompt Prefix of NextlocLLM

- Singapore This data is collected by one mobile SIM card company in Singapore

regarding the spatial representation of trajectories, traditional models like DeepMove typically use location IDs to represent spatial information. In contrast, NextLocLLM utilizes corresponding spatial coordinates to represent positions. This method directly captures spatial relationships, and significantly enhances NextLocLLM's generalization capability in zero-shot tasks. The statistical information for these four datasets can be found in Table 8.

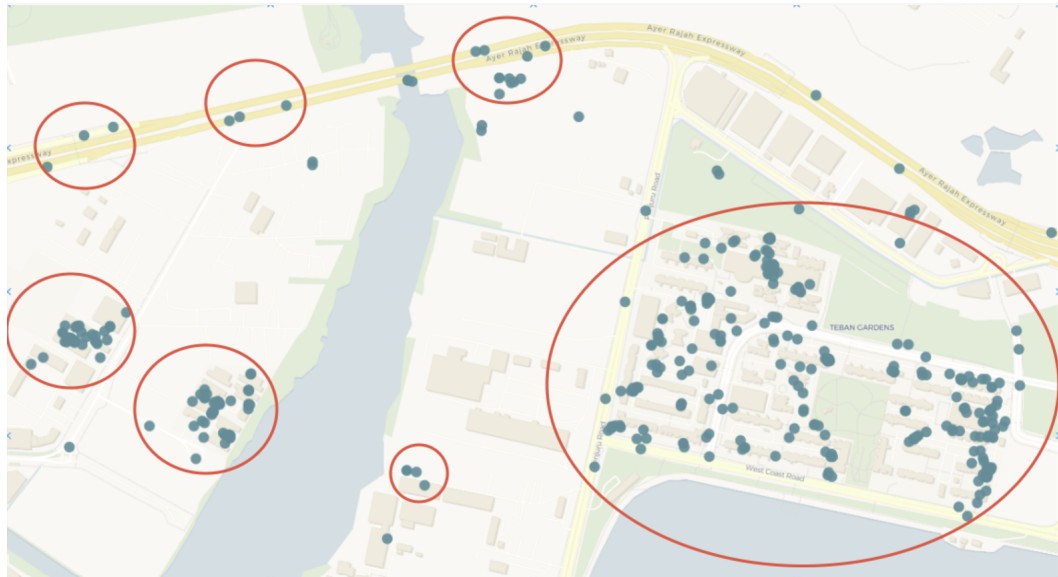

Figure 6: Example of Data processing

To ensure data quality and enhance the model's adaptability to trajectory features, we followed the methodologies of LLM-Mob and C-MHSA and implemented a rigorous preprocessing procedure. The details are as follows:

- Noise Filtering: To effectively filter noise in the raw data, we used the trackintel package to generate staypoints and applied the following filtering rules:
    - Time interval threshold: A staypoint ends when the time interval between consecutive location records exceeds 60 minutes. Subsequent records are treated as a new staypoint.
    - Distance threshold: A new staypoint is generated only if the user's movement exceeds 200 meters, reducing the impact of short-distance fluctuations.
    - Minimum dwell time: A location is considered a valid staypoint only if the user remains there for at least 10 minutes (1 minute for the Singapore dataset).

    These rules effectively removed transient fluctuations and noise from the data, significantly improving data quality.
- Ensuring Trajectory Completeness: To ensure data completeness, we calculated the number of unique days each trajectory spanned (coverage days) and retained only trajectories with

Table 8: Data Statistical Descriptions

| Data Name | Num of Records | Time Span | Num of Users | Avg Interval (minute) | Num of Trajectories |
|-----------|---------------|-----------|--------------|----------------------|---------------------|
| Xian | 23262844 | one year | 690176 | 46.2 | 304870 |
| Chengdu | 22827687 | one year | 712242 | 44.4 | 232021 |
| SG | 900234 | one month | 23461 | 840.35 | 109141 |
| JPN | 88405298 | 60d | 100000 | 95.78 | 2902778 |

more than 15 days of coverage. This criterion helps eliminate short-term noise trajectories and ensures the model learns long-term, stable behavioral patterns, thereby improving robustness and generalization.

- Spatial Grid Partitioning and POI Mapping: Following (Yabe et al., 2024b), we divided the urban area into grids of 500m × 500m and mapped POI data into the corresponding grids. Additionally, we calculated the count of each POI type in each grid. By incorporating the functional characteristics of regions into trajectory features, the model can better capture semantic information and behavioral patterns specific to different areas.

Given that the Singapore dataset originated from unprocessed communication data, we applied extra preprocessing steps prior to the main pipeline:

- Signal delay filtering: Records where the time gap between signal requests and base station reception exceeded 1 minute were removed.
- Low-confidence records: Records with low location confidence were excluded.
- Abnormal elevation: Records with anomalous base station elevations were filtered.
- Geographic bounds: Records outside Singapore's geographic range were discarded.

These additional steps further enhanced the accuracy and usability of the data. As shown in Figure 6, the preprocessing steps effectively reduced noise in the data. The red-circled regions represent the refined staypoints, which are concentrated in meaningful areas such as residential zones, commercial hubs, and transportation nodes. In contrast, the uncircled regions denote noise points that were successfully removed. These results demonstrate that the filtering rules based on time intervals, distance thresholds, and minimum dwell times effectively eliminated spurious or non-meaningful data points, improving the expressiveness of the dataset.

For the Japan dataset, the specific city to which the data belongs is not provided. We created a virtual spatial coordinate system for it, using the geometric center of the dataset as the origin. Specifically, the Japan dataset is distributed across a square spatial grid of 200x200 cells, with the center of the grid at the 100th row and 100th column serving as the origin. This approach does not affect the prediction outcomes because the NextLocLLM model includes a normalization process for spatial coordinate embeddings.

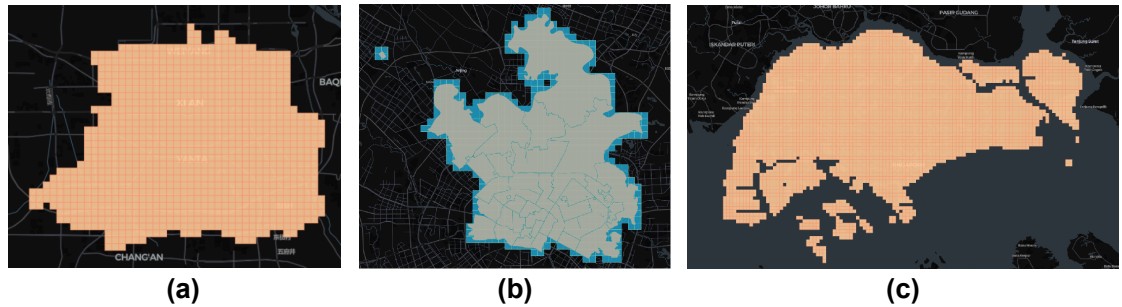

Figure 7: Divided location visualization. (a) Xi'an (b)Chengdu (c)Singapore

## J   POI CATEGORY FREQUENCY CALCULATION

The frequency of POI categories at each location was calculated through the following steps. First, we mapped different types of POIs to their corresponding geographic grids using the geographic

location data recorded in the POI file. This allowed us to identify the specific POI types present within each grid. Next, we counted the frequency of each POI category in every grid, accurately capturing the distribution of POI types across different regions.

To better represent the functional attributes of each area, we aggregated fine-grained POI categories into higher-level functional groups. This aggregation was necessary because individual fine-grained POI categories (e.g., Restaurants, Cafés) often fail to comprehensively reflect the overall functionality of an area. By combining similar or related POI categories, the aggregated functional groups provide a semantically richer and more intuitive representation. For example, a grid containing multiple restaurants and cafés can be classified under the broader functional category of Dining, which directly reflects the primary use of the area without focusing on the specifics of individual POI types.

This aggregation process was based on the classification rules outlined in Tables 6 and 7, which simplify the model's handling of complex fine-grained information. By doing so, we improved the generalizability of the features and enabled the model to efficiently learn patterns related to regional functional attributes. Through these steps, we constructed detailed POI frequency distributions for each location, providing semantically enriched functional attributes that are essential for supporting the model's learning process.

## K  MODEL PARAMETERS AND CONFIGURATIONS

To ensure the reproducibility of the NextLocLLM experiments, we provide a detailed description of the key parameter configurations and input settings below:

**Basic Training Parameters**:

- Learning Rate: 0.0002
- Batch Size: 128
- Max Epochs: 30
- Optimizer: Adam

**Key Hyperparameters**

- LLM Backbone: GPT-2
- Spatial Coordinate Embedding
- Dimension: 128
- Day & Hour Embedding Dimension: 16
- Duration Embedding Dimension: 16
- POI MLP Dimension: 1024
- LLM Layers: 6

**Model Input Configuration**

- Historical Trajectory Length: 40 records, capturing long-term behavioral patterns of users.
- Current Trajectory Length: 5 records, capturing short-term behavioral features.

**LLM Tokenizer and Embedding Layer** We used the tokenizer and embedding layer consistent with the GPT-2 model to ensure compatibility and uniformity in input processing.

**Experimental Hardware** The experiments were conducted on a set of Tesla V100-SXM2-32GB GPUs to support large-scale data training and inference.

## L  FULLY SUPERVISED RESULT FOR SINGAPORE DATASET

In this section, we present the results of NextLocLLM on a private dataset from Singapore, provided by a local mobile signal operator. This dataset is highly sparse, with an average time gap of 840.35

Table 9: Fully Supervised Next Location Prediction Result

| Method | Sinapore | | |
| --- | --- | --- | --- |
| | Hit@1 | Hit@5 | Hit@10 |
| STRNN | 1.073% | 2.652% | 5.269% |
| LSTM | 0.697% | 2.482% | 4.137% |
| FPMC | 0.062% | 0.283% | 0.479% |
| GRU | 1.144% | 2.976% | 4.551% |
| C-MHSA | 1.625% | 5.407% | 8.062% |
| DeepMove | 4.013% | 10.42% | 14.64% |
| GETNext | 2.633% | 7.678% | 10.09% |
| LLMMob(wt) | 4.980% | 15.24% | 22.19% |
| LLMMob(wt,s) | 1.899% | 8.929% | 17.73% |
| LLMMob(wot) | 3.077% | 13.64% | 20.95% |
| LLMMob(wot,s) | 1.763% | 7.671% | 13.42% |
| ZS-NL | 1.077% | 2.399% | 5.064% |
| ZS-NL(s) | 0.958% | 1.926% | 4.401% |
| NextlocLLM | **5.442%** | **17.14%** | **22.36%** |

Table 10: Zero-shot Next Location Prediction Result on Chengdu Dataset

| | Hit@1 | Hit@5 | Hit@10 |
| --- | --- | --- | --- |
| LLMMob(wt) | 45.27% | 81.65% | 84.37% |
| LLMMob(wt,s) | 26.63% | 59.97% | 62.25% |
| LLMMob(wot) | 43.15% | 77.31% | 79.38% |
| LLMMob(wot,s) | 23.37% | 57.71% | 59.36% |
| ZS-NL | 31.06% | 62.25% | 64.47% |
| ZS-NL(s) | 26.78% | 49.57% | 54.67% |
| NextlocLLM(Xi'an->Chengdu) | 61.96% | 96.89% | 98.86% |
| NextlocLLM(Japan->Chengdu) | 54.33% | 88.48% | 91.63% |
| NextlocLLM(Sinapore->Chengdu) | 4.96% | 10.87% | 16.84% |

minutes between consecutive records. Moreover, the location data represents the position of the mobile cell towers rather than the exact locations of the users themselves, introducing additional inaccuracy. These two factors combined contribute to the generally poor performance of all models on this dataset.

Despite these challenges, NextLocLLM outperforms all baseline models, demonstrating its robustness even under difficult conditions. As shown in Table 9, NextLocLLM achieves the highest scores across all key metrics—Hit@1, Hit@5, and Hit@10—outperforming all baseline models.

On the other hand, the limitations of the Singapore dataset, particularly the sparse data and indirect location information, also affect the training quality of the model. As a result, NextLocLLM models trained on this dataset perform suboptimally in zero-shot scenarios, as the parameters cannot be fully optimized during training due to the limited data quality. This lack of sufficient training data on the Singapore dataset results in lower prediction accuracy when applied to other cities in zero-shot tasks.

In summary, while the NextLocLLM model performs better than other baselines on the challenging Singapore dataset, the data sparsity and inaccuracy pose significant challenges, impacting the model's ability to generalize when applied to other datasets in zero-shot scenarios.

# M  ZERO-SHOT RESULT ON OTHER DATASETS

We also compare zero-shot next location prediction performance on other datasets. These experiments share the similar settings with Sec. 5.3. Based on the results in Table 10 and Table!11, NextLocLLM demonstrates a clear advantage over baseline models in zero-shot scenarios across both the Chengdu and Singapore datasets, showing its capablity in cross city generalization.

Table 11: Zero-shot Next Location Prediction Result on Singapore Dataset

| | Hit@1 | Hit@5 | Hit@10 |
|---|---|---|---|
| LLMMob(wt) | 4.981% | 15.24% | 22.19% |
| LLMMob(wt,s) | 1.962% | 8.927% | 17.73% |
| LLMMob(wot) | 1.763% | 7.671% | 13.42% |
| LLMMob(wot,s) | 1.073% | 2.399% | 5.062% |
| ZS-NL | 0.954% | 1.926% | 4.403% |
| ZS-NL(s) | 20.27% | 52.22% | 64.97% |
| NextlocLLM(Xi'an->Singapore) | 5.026% | 16.55% | 19.46% |
| NextlocLLM(Chengdu->Sinapore) | 5.107% | 17.86% | 20.41% |
| NextlocLLM(Japan->Xi'an) | 4.968% | 15.87% | 16.84% |

Table 12: Zero-shot Ablation Study

| Xian-> Chengdu | Hit@1 | Hit@5 | Hit@10 |
|---|---|---|---|
| location | 0% | 0% | 0% |
| coordinate | 61.96% | 96.89% | 98.86% |
| no prompt prefix | 45.84% | 86.11% | 98.84% |

## N PERFORMANCE AMONG DIFFERENT LLMS

Table 13 presents a performance comparison among NextlocLLM using different large language models, including GPT-2, LLama2-7B, and LLama3-8B. We find that among various LLMs, NextlocLLM remains competitive, showing a stable performance among different LLMs.

Our understanding of this phenomenon is as follows. LLMs incorporate a vast amount of information. However, much of this information exceeds the requirements of spatiotemporal sequence analysis. For spatiotemporal prediction tasks, the redundancy of such language-specific information can interfere with the model's ability to effectively capture spatiotemporal patterns, thereby potentially degrading its performance. However, certain aspects of LLMs—particularly those related to human intent and semantic understanding—can significantly enhance spatiotemporal prediction tasks. This duality necessitates a balance when selecting an LLM: minimizing the interference of redundant language information while retaining the semantic understanding capabilities of LLMs to leverage their strengths in modeling human behavioral patterns.

## O COMPARATIVE ANALYSIS OF SPATIAL COORDINATES AND LOCATION IDS

To evaluate the performance of NextLocLLM with different input representations, we conducted experiments using either spatial coordinates or location IDs as inputs among various settings, including fully-supervised and zero-shot scenarios. The results are presented in Table 14. In the fully-supervised setting, models with spatial coordinates inputs consistently outperform those using location IDs, suggesting that coordinate-based representations better capture the spatial relationships between locations, leading to enhanced prediction accuracy. The advantage of coordinate inputs becomes even more pronounced in zero-shot settings, such as Xi'an -> Chengdu or Xi'an -> Singapore. With location ID inputs, the model struggles to generalize across cities, resulting in near-zero Hit@1, Hit@5, and Hit@10 values. This is primarily due to the inconsistency of location ID systems across cities, which lack the semantic or structural information necessary for effective cross-city generalization. In contrast, spatial coordinate inputs enable the model to maintain high performance in zero-shot tasks, with Hit@1 reaching 61.96% in the Xi'an -> Chengdu scenario and 5.026% in

Table 13: Zero-shot Next Location Prediction Result

| | Hit@1 | Hit@5 | Hit@10 | Distance(m) |
|---|---|---|---|---|
| gpt2 | 58.14% | 97.14% | 99.36% | 176.9 |
| Llama2-7B | 50.61% | 84.09% | 96.42% | 290.5 |
| Llama3-8B | 53.29% | 90.66% | 97.45% | 247.6 |

Table 14: Performance comparison between coordinate and location inputs.

| Method | Coordinate | | | Location | | |
|---|---|---|---|---|---|---|
| | Hit@1 | Hit@5 | Hit@10 | Hit@1 | Hit@5 | Hit@10 |
| Xi'an (fully supervised) | 58.14% | 97.14% | 99.36% | 54.57% | 85.11% | 87.88% |
| Singapore (fully supervised) | 5.442% | 17.14% | 22.36% | 2.061% | 11.42% | 15.54% |
| Xi'an $\rightarrow$ Singapore | 5.026% | 16.55% | 19.46% | 0% | 0% | 0% |
| Xi'an $\rightarrow$ Chengdu | 61.96% | 96.89% | 98.86% | 0% | 0% | 0.03% |
| Singapore $\rightarrow$ Xi'an | 3.85% | 9.55% | 19.86% | 0% | 0% | 0.05% |

Xi'an -> Singapore. This demonstrates the superior ability of coordinate-based inputs to generalize spatial knowledge across different urban environments.

## P CASE STUDY

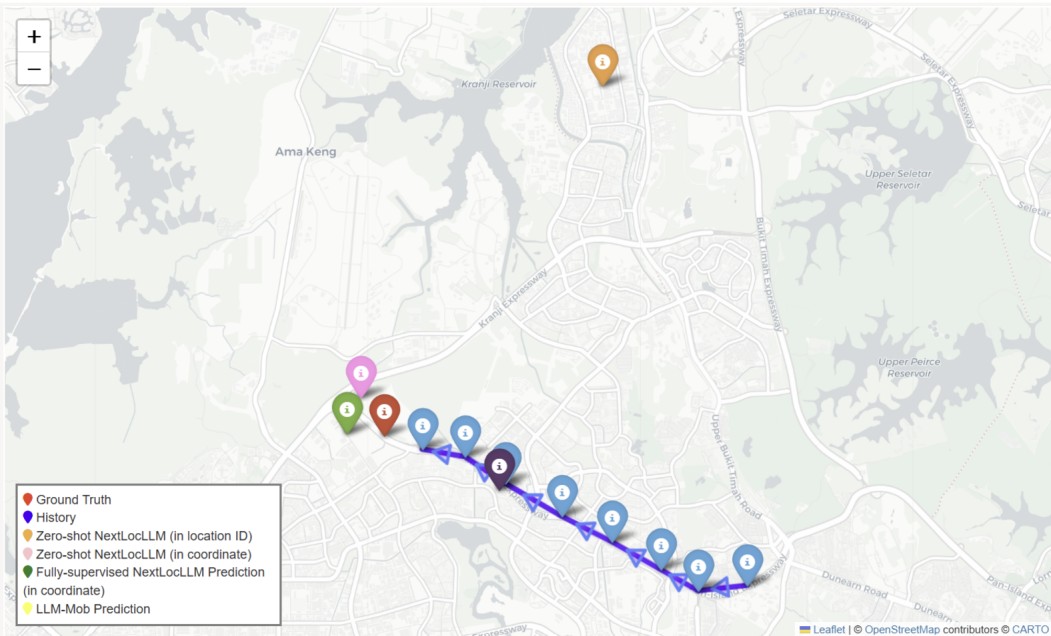

Figure 8: Case study of NextlocLLM

In this case study, we evaluated the zero-shot capabilities of NextLocLLM trained on the Xi'an dataset by testing it on the Singapore dataset. We also compared these results with the fully-supervised setting, where the model was both trained and tested on the Singapore dataset. The Singapore dataset was chosen due to its extensive coverage of location IDs and its distinct POI classification system, which differs from that of Xi'an. This characteristic makes our case study more representative of real-world transfer scenarios, where different cities often use varying POI classification and definition systems, posing a challenge for models to generalize across cities. Specifically, we analyzed a user trajectory from the Singapore dataset with densely sampled points to provide an intuitive demonstration of the model's performance. The trajectory had an average sampling interval of 21 minutes. We compared the following approaches:

- Zero-shot NextLocLLM (using coordinates)
- Zero-shot NextLocLLM (using location IDs)
- Fully-supervised NextLocLLM (using coordinates)
- LLM-Mob prediction

The experimental results clearly highlight the differences in performance across these methods: Zero-shot NextLocLLM (using coordinates) exhibited outstanding predictive capability, generating trajectories highly consistent with the ground truth. This demonstrates NextLocLLM's strong zero-shot generalization in cross-city tasks. In contrast, Zero-shot NextLocLLM (using location IDs) showed significant deviations from the ground truth. This result underscores the limitations of location ID-based models in cross-city tasks, as inconsistent location ID systems across cities fail to convey the semantic information of geographic spaces. LLM-Mob prediction consistently limited its forecasting to locations explicitly mentioned in the prompts, revealing that models purely based on prompts and location IDs struggle to comprehend spatial information or capture trajectory sequence patterns. This limitation significantly reduces the applicability of LLM-Mob for tasks involving unknown locations, particularly in zero-shot cross-city scenarios. Finally, Fully-supervised NextLocLLM (using coordinates) set the upper performance bound in a supervised setting, achieving results only marginally better than the zero-shot setting. This further validates the robust generalization capability of NextLocLLM, which maintains near-supervised performance even without target city training data.

## Q  HYPERPARAMETER SENSITIVITY

We conducted a detailed sensitivity analysis of the key hyperparameters to validate the robustness of NextLocLLM's performance. Below is an analysis of the experimental results: **Relatively Stable Parameters:** $d_{xy}$**,** $d_t$**,** $d_d$**,** $d_{poi}$. These parameters exhibit relatively smooth effects on the model's performance:

- *Spatial Coordinate Embedding Dimension ($d_{xy}$):* As the dimension increases from 64 to 256, the model's performance improves steadily. However, beyond 256, the performance saturates, indicating that smaller embedding dimensions are sufficient to capture geographic information effectively.
- *Stay Duration Embedding Dimension ($d_t$):* Performance shows slight improvement as the dimension increases from 8 to 32, but further increases have minimal impact, suggesting that the embedding requirements for stay duration information are relatively low.
- *Date and Time Embedding Dimension ($d_d$):* Hit@1 performance remains stable as the dimension increases from 8 to 32, demonstrating that date and time features have a minimal yet robust effect on model performance.
- *POI Embedding Dimension ($d_{poi}$):* As the dimension increases from 256 to 1024, performance improves significantly, but stabilizes beyond 1024, indicating that overly large dimensions may introduce redundant information.

In summary, variations in these parameters within a reasonable range have limited impact on the model's performance, highlighting the robustness of the model to these hyperparameter settings.

**Parameters with Greater Impact: Historical Trajectory Length ($M$) and LLM Layers ($N_{layers}$):**

- *Historical Trajectory Length ($M$):* As the historical trajectory length increases, performance steadily improves, suggesting that incorporating more historical information helps the model better understand past travel patterns and preferences.
- *LLM Layers ($N_{layers}$):* Performance improves significantly as the number of layers increases from 3 to 12, indicating that deeper architectures enhance the model's representational capacity. However, beyond 6 layers, the performance stabilizes. Considering the trade-off between efficiency and performance, we chose to use a 6-layer GPT configuration.

This sensitivity analysis demonstrates the robustness of the model to most parameter settings while identifying key parameters that have a more pronounced effect on performance.

## R  TRAINING AND INFERENCE TIME

We conducted an in-depth analysis and supplementary experiments focusing on training and inference times. The results highlight key insights into NextLocLLM's performance.

Table 15: Training time comparison

| Model | Training Time (s/iter) |
|---|---|
| GRU | 0.131 |
| DeepMove | 0.155 |
| c-MHSA | 0.199 |
| NextLocLLM | 0.935 |

Table 16: Inference time comparison

| Model | Inference Time (s) |
|---|---|
| NextLocLLM | 229.2 |
| LLM-Mob | 95232 |

For training, the per-iteration training time of NextLocLLM is 0.935 seconds, which is higher compared to GRU (0.131 seconds), DeepMove (0.155 seconds), and c-MHSA (0.199 seconds). However, during training, NextLocLLM significantly reduces the computational burden by freezing most of the LLM backbone parameters and only optimizing a small number of newly added module parameters. This design not only reduces the number of trainable parameters but also accelerates model convergence. For instance, on the Xi'an dataset, NextLocLLM converges in approximately 15 epochs. As a result, the total training time is manageable, and the substantial performance improvement achieved by the model justifies the slightly higher computational cost.

In terms of inference efficiency, NextLocLLM demonstrates a clear advantage. For example, on the Xi'an dataset, its inference time is 229.2 seconds, which is significantly lower than LLM-Mob's 95,232 seconds. This efficiency gain can be attributed to two key factors. First, NextLocLLM supports batch inference, allowing it to predict multiple trajectories simultaneously, in contrast to LLM-Mob and ZS-NL, which rely on prompt engineering and process trajectories sequentially. This parallelized design greatly reduces the total inference time. Second, the POI functional attributes for each location remain consistent across trajectory records. In NextLocLLM, the LLM-enhanced POI embeddings are precomputed once during data loading and stored for reuse, eliminating the need for repeated calculations during inference and substantially reducing the computational overhead.

We acknowledge that there is room for improvement in NextLocLLM's computational efficiency. In future work, we plan to further optimize the model by exploring techniques such as quantization and pruning to reduce computational overhead through optimized model weights and structures. Additionally, we aim to investigate lightweight model designs through distillation methods to further reduce the model size and enhance real-time performance.

## S    ANALYSIS OF ZERO-SHOT CAPABILITY

To provide a comprehensive explanation of the model's performance in zero-shot scenarios, we conducted an in-depth analysis of the key design components, particularly their contributions to cross-city tasks, and validated their effectiveness through experiments.

First, we examined the relationship between coordinate-based inputs and zero-shot performance. In the Xi'an $\rightarrow$ Chengdu experiment, when using location IDs as inputs, the model performed poorly in the zero-shot setting, with Hit@1, Hit@5, and Hit@10 scores nearly at 0%. This indicates that location IDs cannot generalize across cities due to the lack of consistency in ID systems, which fail to convey structural information inherent in geographic spaces. In contrast, when spatial coordinates were used as inputs, the model's performance in the zero-shot scenario improved significantly, achieving a Hit@1 of 61.96%, Hit@5 of 96.89%, and Hit@10 of 98.86%. This demonstrates that spatial coordinates provide a universal representation capable of effectively capturing geographic relationships, thereby enhancing the model's generalization capability for cross-city tasks.

Second, we investigated the role of prompt prefixs in the model's zero-shot ability. When prompt prefixs were removed, the model's performance dropped noticeably, with Hit@1 decreasing from 61.96% to 45.84%. This result highlights the importance of prompt prefixs in helping the model understand data characteristics and task objectives. Prompt prefixs provide clear contextual infor-

Table 17: Training time comparison

| Model | Training Time (s/iter) |
|-------|------------------------|
| GRU | 0.131 |
| DeepMove | 0.155 |
| c-MHSA | 0.199 |
| NextLocLLM | 0.935 |

Table 18: Usage of LLM (fully-supervised scenario)

| Fully-supervised(Xi'an) | Hit@1 | Hit @5 | HIt @10 |
|-------------------------|-------|--------|---------|
| NextLocLLM | 58.14% | 97.14% | 99.36% |
| LLM->Transformer | 45.62% | 83.25% | 88.78% |

mation that guides the model to integrate trajectory data, POI embeddings, and other features more effectively, thereby enhancing its generalization capabilities.

These results clearly demonstrate that NextLocLLM's zero-shot capability stems from its use of spatial coordinates, which provide universal geographic information, and prompt prefixs, which enhance the model's understanding of the task context and objectives. Together, these innovations enable NextLocLLM to outperform other LLM-based methods in cross-city tasks.

## T    USAGE OF LLM

The primary advantage of integrating LLMs into this task lies in their ability to transcend the limitations of small models. Traditional small models typically rely on statistical patterns extracted from large trajectory datasets to predict the next location. While effective in data-rich scenarios, these models often overlook the semantic nature of trajectories—the underlying human behavioral patterns reflected in the real world. In data-scarce or zero-shot scenarios, where statistical patterns alone are insufficient, trajectory semantics become critical. LLMs, with their powerful natural language understanding and reasoning capabilities, excel in capturing the inherent semantics of trajectories, even with limited data. This is the foundational motivation behind using LLMs as the core design of our framework.

To maximize the utility of LLMs, we optimized our model to enhance its semantic understanding of trajectories. First, we employed prompt prefixes to clearly define the prediction task, coupled with natural language descriptions of POIs to convey their functional attributes. This design leverages the LLM's ability to interpret natural language and human behavioral patterns, enabling it to go beyond learning statistical patterns and uncover the deeper semantic and logical relationships within trajectories. This capability is particularly beneficial for transfer tasks, where such semantic understanding significantly boosts performance.

To validate our hypothesis, we conducted experiments where we replaced the LLM in our framework with a randomly initialized Transformer model and trained it on the same Xi'an dataset. The results showed that this model performed significantly worse in zero-shot and transfer tasks, with a particularly notable gap in zero-shot scenarios. This demonstrates the importance of LLM's pre-trained knowledge and reasoning capabilities. The LLM effectively utilizes its internal semantic and logical information to complement the lack of statistical data. Even when the Transformer model was trained on a large amount of data, its performance still fell short of the LLM, further confirming the LLM's unique strengths in transfer tasks.

It is important to note that in fully supervised scenarios with abundant data, small models may hold certain advantages, such as computational efficiency and faster training times. However, in real-world applications, data scarcity and transfer tasks are common challenges. In these scenarios, the LLM's ability to understand and reason about the semantics of trajectories gives it a clear edge.

Table 19: Usage of LLM (Zero-shot scenario)

| Fully-supervised(Xi'an->Chengdu) | Hit@1 | Hit @5 | HIt @10 |
|---|---|---|---|
| NextLocLLM | 61.96% | 96.89% | 98.86% |
| LLM->Transformer | 31.18% | 51.65% | 62.71% |

| Grid Resolution | C-MHSA | | | NextLocLLM | | |
|---|---|---|---|---|---|---|
| | **Hit@1** | **Hit@5** | **Hit@10** | **Hit@1** | **Hit@5** | **Hit@10** |
| 500m*500m | 50.32% | 92.43% | 95.38% | 58.14% | 97.14% | 99.36% |
| 200m*200m | 40.63% | 82.26% | 86.77% | 50.99% | 88.42% | 94.77% |
| 50*50 | OOM | OOM | OOM | 36.87% | 62.95% | 72.47% |

## U  INFLUENCE OF GRID RESOLUTION

We conducted experiments on Xi'an to evaluate the performance of NextLocLLM and C-MHSA under different grid sizes (500m×500m, 250m×250m, and 50m×50m) and analyzed how grid resolution affects the models' performance.

The experimental results show that for C-MHSA, finer grid resolutions lead to a noticeable decline in performance. When the grid size is reduced to 50m×50m, the number of candidate locations increases substantially, resulting in significantly higher computational overhead and eventually causing an out-of-memory (OOM) error. In contrast, NextLocLLM, which directly predicts coordinates without relying on grid partitions, successfully completes predictions even under fine-grained grid settings. While its performance on metrics such as Hit@1 declines at higher resolutions, it still outperforms C-MHSA overall, demonstrating robust adaptability to changes in spatial resolution.

This experiment also highlights some areas for potential improvement in NextLocLLM when handling high-resolution prediction tasks. For instance, under extremely fine grid settings (e.g., 50m×50m), the model's performance still has room for enhancement, suggesting that its ability to model coordinate precision could be further improved.

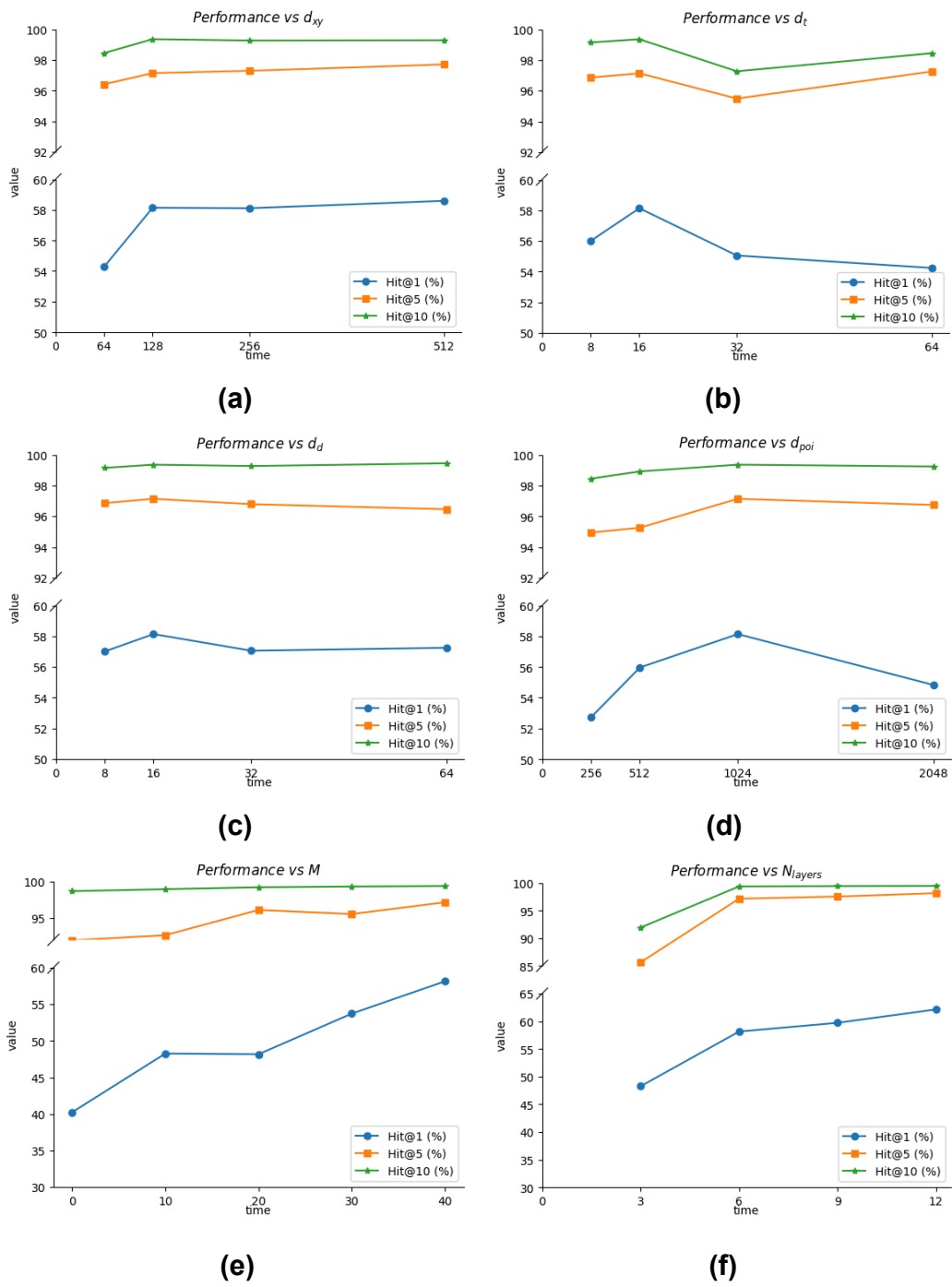

Figure 9: Hyperparameter sensitivity analysis

