# OpenReview forum: "NEXTLOCLLM: NEXT LOCATION PREDICTION USING LLMS"
_ICLR.cc/2025/Conference — ICLR 2025 Conference Desk Rejected Submission_

### Official Review · Reviewer_arN2 · 2024-10-29

**Soundness:** 2
**Presentation:** 2
**Contribution:** 2
**Rating:** 5
**Confidence:** 4

**Summary:**

This paper addresses a significant research question concerning model transferability across different cities and introduces NextLocLLM, which integrates large language models (LLMs) with next location prediction models. The method incorporates multi-dimensional trajectory content embeddings, LLM-enhanced POI embedding, an LLM backbone, and a prediction retrieval module, purportedly achieving state-of-the-art (SOTA) results in both fully supervised and zero-shot settings. Despite these claims, the experimental design, framework rationale, and methodological clarity are not convincing enough, leading me to lean towards rejection.

**Strengths:**

1. The paper tackles the crucial issue of model transferability across urban settings, a topic of growing importance in location-based services.
2. This paper demonstrates improvements in performance, achieving state-of-the-art results in both fully supervised and zero-shot next location predictions.
3. This paper innovatively integrate multiple data sources, including points of interest (POI), textual descriptions, and trajectory data, enriching the model’s contextual understanding.

**Weaknesses:**

1. The experiments are not solid. For example, 1) why using LLaMA2 and LLaMA3 does not outperform GPT2. 2) It seems that the proposed KD-Tree-based prediction contributes, but you did not involve it in the experiments. 3) In the ablation study, only using LLM-enhanced POI achieves the second-best result. It challenges the significance of the proposed methods. In addition, it is weird that using both POI and LORA will lead to worse performance than pure POI.

2. This framework seems not reasonable. This work aims to achieve generalization ability by LLM. However, I am wondering whether this ability can be achieved without road graph. It seems that this paper aim to realize zero-shot generalization with purely POI even without trajectory.

3. The formalization is a little bit confusing. TI=\{(d,t)\} and t is time-of-day (0 ≤ t ≤ 23 in hours). Since Taxi data seem to be less than an hour. This formalization seems not complete.

**Questions:**

Please explain the weaknesses.

---

> ### Author Response · Authors · 2024-11-20
> **Response to Reviewer arN2 (1/3)**
>
> > W1  why using LLaMA2 and LLaMA3 does not outperform GPT2.
>
> This is a profound question that touches on the core of how LLMs handle spatiotemporal problems. Previous studies have also observed similar phenomena. For instance, [1] found that GPT-2 outperformed LLaMA2, attributing this to GPT-2’s parameter scale being better suited for relatively smaller datasets. Similarly, in [2], experiments demonstrated that GPT-2 achieved better results than LLaMA3 in next location prediction tasks. These findings collectively suggest that smaller-scale LLMs are often better suited for spatiotemporal prediction problems.
>
> Our understanding of this phenomenon is as follows. LLMs contain a vast amount of knowledge from text corpus, while much of the knowledge  exceeds the requirements of spatiotemporal sequence analysis. For spatiotemporal prediction tasks, the redundancy of such language-specific information can interfere with the model’s ability to effectively capture spatiotemporal patterns, thereby potentially degrading its performance. However, certain aspects of LLMs—particularly those related to human intent and semantic understanding—can significantly enhance spatiotemporal prediction tasks.  This duality necessitates a balance when selecting an LLM: minimizing the interference of redundant language information while retaining the semantic understanding capabilities of LLMs to leverage their strengths in modeling human behavioral patterns.
>
> Based on this understanding, we selected GPT-2 as the core architecture for NextLocLLM, rather than larger-scale models like LLaMA2 or LLaMA3. GPT-2’s smaller parameter size and more streamlined training corpus make it better aligned with the demands of spatiotemporal prediction tasks, while still preserving its core strengths in natural language understanding. Our experimental results support this choice: under identical experimental settings, GPT-2 outperformed LLaMA2 and LLaMA3, demonstrating its suitability for this specific task context. We will provide further related analysis in the revised manuscript. Thank you again for your valuable feedback!
>
>
>
> [1] Cheng J, Yang C, Cai W, et al. NuwaTS: Mending Every Incomplete Time Series[J]. arXiv preprint arXiv:2405.15317, 2024.
>
> [2]  Wang X, Fang M, Zeng Z, et al. Where would i go next? large language models as human mobility predictors[J]. arXiv preprint arXiv:2308.15197, 2023.

---

> ### Author Response · Authors · 2024-11-20
> **Response to Reviewer arN2 (2/3)**
>
> > W2 It seems that the proposed KD-Tree-based prediction contributes, but you did not involve it in the experiments.
>
> We would like to clarify that our contribution lies beyond KD-tree. It lies in introducing a regression-based approach to replace traditional classification methods, providing a more effective solution for zero-shot next location prediction tasks. The KD-tree plays a supportive role to accelerate the post-processing of regression outputs. We will revise our manuscript to make this distinction clearer.
>
> Traditionally, next location prediction tasks are formulated as classification problems, where each possible location ID is treated as a class, and the model predicts a probability distribution over these classes to determine the next location. However, this approach has significant limitations in zero-shot scenarios. Since location ID systems differ completely between cities, classification-based methods fail to transfer effectively.
>
>  In contrast, NextLocLLM employs a regression-based approach that directly predicts the spatial coordinates of the next location instead of discrete class labels. This design effectively addresses the limitations of traditional classification methods, removing the need for unified mappings across cities and significantly enhancing transferability. Additionally, regression predicts precise latitude and longitude coordinates, offering superior geographic resolution compared to classification, which is restricted to discrete grids or predefined locations..
>
> To address your concern, we have further conducted experiments to test the contribution of the use of KD-tree when not considering the zero-shot scenario. Specifically, after predicting spatial coordinates (x,y), we replace the KD-tree nearest-neighbor search with a linear layer and softmax function in the output layer, redefining the task as a classification problem that outputs a probability distribution over all possible locations with cross-entropy loss.
>
> The results show that in fully supervised scenarios, KD Tree consistently outperforms the classification-based approach across all three metrics. This reinforces the advantages of the regression-based framework in both zero-shot and fully supervised settings, demonstrating its flexibility and robustness.
>
> In summary, our innovation lies in the regression-based design, with KD-tree enhancing its efficiency. We will revise the manuscript to clarify these points, include the supplementary experimental results, and further detail the role of KD-tree in our framework. Thank you again for your valuable feedback!
>
> | Method               | Hit@1   | Hit@5   | Hit@10  |
> |----------------------|---------|---------|---------|
> | KD TREE             | 61.96%  | 96.89%  | 98.86%  |
> | Linear + Softmax Layer | 49.80%  | 70.86%  | 77.52%  |

---

> ### Author Response · Authors · 2024-11-20
> **Response to Reviewer arN2 (3/3)**
>
> > W3.1   In the ablation study, only using LLM-enhanced POI achieves the second-best result. It challenges the significance of the proposed methods.
>
> We would like to first clarify that 'only using LLM-enhanced POI' in this ablation study is equivalent to 'not using prompt prefix'. Indeed, ' only using LLM-enhanced POI embedding' performs exceptionally well in the ablation study, second only to the complete model. However, we believe this result  actually demonstrates the significance and validates the rationale and unique advantages of our approach.
>
> We argue that one main advantage of LLMs in  next location prediction  lies in their semantic understanding capabilities. By analyzing POI information, LLMs can better understand the functional attributes of locations, allowing them to infer the human behavioral intentions hidden in the trajectories. This is something that traditional small models struggle to achieve. Even when using only LLM-enhanced POI embeddings, LLMs achieve high performance, which demonstrates that their understanding of regional semantics contributes significantly to the model's performance.
>
> > W3.2 In addition, it is weird that using both POI and LORA will lead to worse performance than pure POI.
>
> As for why the introduction of LoRA results in a performance drop, this is a noteworthy issue in the field of spatiotemporal LLMs. A similar phenomenon was observed in [1]，where the study pointed out that QLoRA's low-rank updates could lead to information loss when handling complex semantic features.
>
> For this phenomenon, we hypothesize that LLMs, trained on large-scale textual corpora, have absorbed rich language-specific knowledge, much of which goes beyond the requirements of spatiotemporal sequence analysis. In spatiotemporal prediction tasks, these language-related details may interfere with the model’s ability to capture spatiotemporal patterns effectively, resulting in performance degradation. LoRA fine-tuning, which updates the model’s parameters, might further activate or amplify these irrelevant language-specific features, exacerbating the interference and leading to diminished prediction accuracy.  We hope our response answers your questions.
>
> [1]Jin M, Wang S, Ma L, et al. Time-LLM: Time Series Forecasting by Reprogramming Large Language Models[C]//The Twelfth International Conference on Learning Representations.
>
> >W4 This work aims to achieve generalization ability by LLM. However, I am wondering whether this ability can be achieved without road graph. It seems that this paper aim to realize zero-shot generalization with purely POI even without trajectory.
>
> We would like to clarify that our study specifically targets the prediction of **mobility with semantic patterns** (e.g., stay point as locations) rather than road network-constrained vehicle trajectory prediction.
>
>  Specifically, our method leverages POI data to represent functional attributes (e.g., commercial, residential or educational areas). This representation allows for cross-city generalization, enabling the model to align functionally similar regions between cities, thereby significantly enhancing performance in zero-shot transfer tasks.
>
> Moreover, in our method, **historical and current trajectories remain the core data sources**, capturing both short-term and long-term movement patterns of users. These trajectory records provide critical dynamic movement information and serve as the primary basis for predicting the next location. POI information plays a complementary role by enhancing the model’s understanding of regional characteristics, providing semantic support for functional areas. This design allows the model to analyze user movement patterns while leveraging regional functionality to make more informed predictions, articularly in cross-city scenarios.
>
> > W5  The formalization is a little bit confusing. TI={(d,t)} and t is time-of-day (0 ≤ t ≤ 23 in hours). Since Taxi data seem to be less than an hour. This formalization seems not complete
>
>
> In our task, the time feature $t$ is defined as the hour of the day, with a range from 0 to 23 (i.e., time-of-day). For trajectory data with sampling intervals smaller than an hour, all trajectory records falling within the same hour are assigned the same hour-level time feature. This approach is based on the assumption that trajectory records within the same hour generally exhibit similar behavioral patterns and are not significantly affected by minute-level variations. For example, trajectory records at 10:15 and 10:45 are, in most cases, considered to share similar temporal feature. This processing method avoids overly fine-grained time segmentation, thereby reducing the complexity of learning time features while still capturing the primary temporal characteristics of different behavioral patterns.

---

> ### Comment · Reviewer_arN2 · 2024-11-22
>
> Thank you for your response. While it addressed some of my concerns, due to the following reasons, I will maintain my score.
>
> 1) The primary motivation of the paper is to leverage large language models (LLMs) to enhance next-location prediction. However, the experimental results suggest that better-performing LLMs such as LLaMA do not necessarily lead to improved prediction outcomes. This raises questions about the suitability and effectiveness of the proposed framework for the stated task.
>
> 2) Although the paper reports state-of-the-art performance in Table 1, it is unclear whether this improvement stems from the proposed framework itself or primarily from the use of the KD tree. This distinction warrants further investigation to validate the novelty and contribution of the framework.
>
> 3) The clarity and precision of the manuscript require further refinement. For instance, in Definition 2.4, the authors state that the goal is to predict the next location $loc_{t+1}$. However $loc$ is defined as a tuple $ (id,x,y,poi)$. It seems that the prediction target is the $id$ rather than the entire tuple. This ambiguity could lead to misinterpretations and should be clarified.
>
> 4) The experiments are conducted on a taxi dataset, which typically involves trips with predefined destinations. I question the necessity of including POI information in this context, as the POIs along the route are merely passed by and may not hold significant relevance to the prediction task. This design choice should be better justified, or its impact should be clearly demonstrated.

---

> ### Author Response · Authors · 2024-11-22
> **Additional Response to Reviewer arN2 (1/3)**
>
> Thank you for your timely feedback. We will provide a more detailed explanation here regarding the new issues you raised. We hope this will address your concerns and enable you to reassess our work.
>
> > Additional Q1 The primary motivation of the paper is to leverage large language models (LLMs) to enhance next-location prediction. However, the experimental results suggest that better-performing LLMs such as LLaMA do not necessarily lead to improved prediction outcomes. This raises questions about the suitability and effectiveness of the proposed framework for the stated task.
>
> Firstly, from the perspective of experimental results, we compare NextLocLLM using LLaMA2, LLaMA3, and GPT2 as its backbone against other baselines. Regardless of whether in fully-supervised or zero-shot scenarios, NextLocLLM demonstrates advantages. In fully-supervised scenarios, NextLocLLM consistently outperforms traditional small models in most cases. Moreover, no matter which LLM is used as the backbone, NextLocLLM achieves superior performance compared to other LLM-based methods in zero-shot scenarios, where traditional small models are unable to generalize across cities. This indicates that our design framework of NextLocLLM and its integration with LLMs are both effective and meaningful.
>
> Secondly, from the perspective of theoretical analysis, *while LLaMA generally outperforms GPT2 in most natural language-related tasks, this does not necessarily imply it is superior in all non-textual tasks*. In fact, numerous studies have found that in certain specific non-textual tasks, LLaMA as a backbone performs worse than GPT2. For example, [1][2][3][4][5][6] have observed such phenomena. Among them, [4] suggests that this may be due to the parameter scale of LLaMA being overly redundant for the requirements of time-series tasks. Combined with our analysis in response to W1, we suppose that LLaMA contains more redundant information learned from text corpora compared to GPT2. While such information is beneficial for natural language processing tasks, it may interfere with the model's ability to effectively capture spatiotemporal patterns and trajectory semantic information in spatiotemporal tasks, thereby affecting prediction performance.
>
> The above summarizes our understanding of this issue. We hope that the analysis from both experimental and theoretical perspectives can address your concerns.
>
> | Method (Xian fully-supervised)                       | Hit@1  | Hit@5  | Hit@10 |
> |------------------------------|--------|--------|--------|
> | C-MHSA                      | 50.32% | 92.43% | 95.38% |
> | DeepMove                    | 41.19% | 83.02% | 90.85% |
> | GETNext                     | 48.63% | 85.67% | 93.25% |
> | NextLocLLM (GPT-2)          | 58.14% | 97.14% | 99.36% |
> | NextLocLLM (Llama2-7B)      | 50.61% | 84.09% | 96.42% |
> | NextLocLLM (Llama3-8B)      | 53.29% | 90.66% | 97.45% |
>
> | Method(Xian->Chengdu)                       | Hit@1  | Hit@5  | Hit@10 |
> |------------------------------|--------|--------|--------|
> | LLMMob (wt)                 | 45.27% | 81.65% | 84.37% |
> | LLMMob (wot)                | 43.15% | 77.31% | 79.38% |
> | ZS-NL                       | 31.06% | 62.25% | 64.47% |
> | NextLocLLM (GPT-2)          | 61.96% | 96.89% | 98.86% |
> | NextLocLLM (Llama2-7B)      | 53.32% | 84.94% | 91.32% |
> | NextLocLLM (Llama3-8B)      | 55.79% | 88.19% | 94.77% |
>
> [1] Tempo: Prompt-Based Generative Pre-Trained Transformer for Time Series Forecasting
>
> [2] AutoTimes: Autoregressive Time Series Forecasters via Large Language Model
>
> [3] Enhancing Graph Neural Networks with Limited Labeled Data by Actively Distilling Knowledge from Large Language Model
>
> [4] NuwaTS: Mending Every Incomplete Time Series
>
> [5]  Where would i go next? large language models as human mobility predictors
>
> [6] Spatial-Temporal Large Language Model for Traffic Prediction

---

> ### Author Response · Authors · 2024-11-22
> **Additional Response to Reviewer arN2 (2/3)**
>
> > Additional Q2 Although the paper reports state-of-the-art performance in Table 1, it is unclear whether this improvement stems from the proposed framework itself or primarily from the use of the KD tree. This distinction warrants further investigation to validate the novelty and contribution of the framework.
>
> In addition to our explanation of the role of the KD tree in response to W2, we believe that the right-hand section of Table 13 provides further critical evidence to address your concerns. In this part of the experiment, we calculate the average distance between the predicted coordinates from NextLocLLM and the actual coordinates when using different LLMs (e.g., GPT-2, LLaMA2, and LLaMA3) as the backbone. Furthermore, we conduct an additional experiment where the input and output of C-MHSA—the best-performing small model in the fully-supervised scenario—are modified from discrete location IDs to spatial coordinates, consistent with NextLocLLM. The average distance error of its predictions is also calculated. This prediction of spatial coordinates occurs before the KD tree query and directly reflects the effectiveness of the framework itself, independent of the KD tree’s impact.
>
> The experimental results show that regardless of the specific LLM backbone used, the average distance between the predicted and actual positions for NextLocLLM is significantly smaller than that of C-MHSA. Additionally, the prediction errors of NextLocLLM with different backbones are generally less than half the size of a location grid. This indicates that most predicted coordinates fall within the location of the ground truth, meaning the predictions are correct. This demonstrates that the performance improvements of our proposed framework are significant, and the KD tree's role is to mapping the predicted coordinates to the nearest top-k locations. If the model itself has high prediction error, even an efficient KD tree query cannot compensate for the discrepancy between the predicted coordinates and the actual positions.
>
> Therefore, we can confidently conclude that the core source of NextLocLLM's performance improvement lies in our proposed framework design itself, rather than the KD tree. We hope this additional experiment and analysis further address your concerns!
>
> | Xian                         | Distance (m) |
> |------------------------------|--------------|
> | C-MHSA (reg)                | 588.2        |
> | NextLocLLM (GPT-2)          | 176.9        |
> | NextLocLLM (Llama2-7B)      | 290.5        |
> | NextLocLLM (Llama3-8B)      | 247.6        |
>
>
> > Additional Q3 The clarity and precision of the manuscript require further refinement.
>
> Thank you for your valuable feedback! In response to the issue you raised, we have revised Definition 2.4 to explicitly clarify that the prediction target is the ID of the next location, thereby avoiding any ambiguity that might lead to interpreting the target as the entire tuple. This revision further ensures the clarity and accuracy of the manuscript, minimizing the potential for misunderstanding. We sincerely appreciate your suggestion!

---

> ### Author Response · Authors · 2024-11-22
> **Additional Response to Reviewer arN2 (3/3)**
>
> >Additional Q4  The experiments are conducted on a taxi dataset, which typically involves trips with predefined destinations. I question the necessity of including POI information in this context, as the POIs along the route are merely passed by and may not hold significant relevance to the prediction task. This design choice should be better justified, or its impact should be clearly demonstrated.
>
> On this point, we will explore the role and importance of POI information in this task from three perspectives: data processing, method design, and its contribution to cross-city tasks.
>
> First, in terms of data preprocessing, we follow the approach adopted in [1][2] to preprocess the trajectory data of the Xi’an and Chengdu datasets, converting them into staypoints to eliminate transit trajectory points and focus on locations where users remain relatively stationary. Specifically, we set two key thresholds: a minimum stay duration of 10 minutes and a distance of more than 200 meters from the previous staypoint. For the Singapore dataset, due to its characteristics as population mobility data, we adjust the stay duration threshold to 1 minute. This preprocessing method effectively filters out transient points and retains only staypoints relevant to the prediction task. These staypoints are then mapped to specific locations and serve as input data for our next location prediction task.
>
> Second, in terms of method design, introducing POI information aims to enhance the model's understanding of the functional attributes of locations, thereby improving its ability to capture the semantic information behind user behavior. Spatial coordinates only reflect the geographical characteristics of a location, while POI information reveals its functional properties, such as whether the location is a commercial area, a residential area, or a transportation hub. Such functional information is crucial for predicting user behavior and preferences. Even in taxi trajectory data, drivers exhibit preferences, such as whether they tend to stay in commercial areas or transportation hubs when not carrying passengers. Incorporating POI information as a supplement to spatial semantics and conducting experiments on vehicle trajectory data **is not uncommon** in next location prediction tasks. For instance, [1] integrated POI information using an LDA model on the Swiss Green Class Car dataset, and [3] normalized POI counts for experiments on taxi datasets in Shanghai, Chengdu, and San Francisco. These studies demonstrate the effectiveness of POI information in enhancing the model's understanding of the functional properties of vehicle trajectory locations. Compared to these methods, our proposed LLM-enhanced POI embedding further improves the understanding of POI information by capturing its deeper functional properties and providing a unified approach to modeling the semantic functions of locations. This approach not only overcomes the limitations of fixed POI taxonomy but also offers stronger generalization capabilities, particularly excelling in cross-city tasks.
>
> Finally, in cross-city prediction scenarios, POI information is especially important for understanding the spatial semantics of locations. Solely relying on spatial coordinates for modeling has significant limitations in cross-city tasks. For example, in the Xi’an dataset, major commercial areas are concentrated in the city center, while in the Singapore dataset, commercial areas are distributed not only in the city center but also in numerous suburban satellite towns. If modeling relies only on spatial coordinates, it becomes difficult for the model to capture the similarities and differences in the functional attributes of  locations in different cities. However, with the introduction of POI information, our model can better align locations with similar functional attributes across different cities, thereby improving generalization ability. To validate this, we conduct additional ablation experiments, and the results shows that removing POI information significantly reduces prediction performance, further demonstrating its critical role in this task.
>
> || With POI||| Without POI  | ||
> |--------------------|-----------------------|--------------------------|--------------------------|-------------------------|--------------------------|--------------------------|
> | | Hit@1| Hit@5| Hit@10| Hit@1| Hit@5| Hit@10|
> | Xi'an -> Singapore | 5.026%| 16.55%| 19.46%| 1.422%                 | 9.967%                  | 10.232%                 |
> | Xi'an -> Chengdu   | 61.96%| 96.89%| 98.86%                  | 42.27%                 | 81.32%                  | 89.92%                  |
>
> [1] Context-aware multi-head self-attentional neural network model for next location prediction
>
> [2]Where would i go next? large language models as human mobility predictors
>
> [3] An Efficient Destination Prediction Approach Based on Future Trajectory Prediction and Transition Matrix Optimization

---

> > ### Comment · Reviewer_arN2 · 2024-11-23
> >
> > Thank you for your response. I appreciate the clarification provided regarding the dataset, which addressed my concerns on that front. However, the experimental results involving LLaMA and GPT still raise questions about the fundamental rationale for utilizing LLMs in this context.
> >
> > Therefore, I acknowledge and value the significant effort you have made in conducting experiments and providing illustrations to support your work. In light of this, I will raise my score to 5.

---

> > > ### Author Response · Authors · 2024-11-24
> > > **Further Response to Reviewer arN2(1/2)**
> > >
> > > Thank you for your feedback and recognition of our work! We are delighted to engage in deeper academic discussions with you. To address your concerns about the rationale for incorporating large language models (LLMs) in the next location prediction task, we will analyze the inherent requirements of this task from a new perspective. We will first identify the limitations of traditional small models in handling this task and then demonstrate how our model leverages LLMs as data enhancers and predictors to address these challenges. Subsequently, we will present experimental results to support our argument that the rich pre-trained knowledge and reasoning capabilities of LLMs significantly enhance next location prediction performance. Furthermore, we will analyze experimental results involving LLaMA and GPT, offering explanations of how LLMs contribute to task improvements.
> > >
> > > Trajectory data encapsulates abundant patterns of human behavior and semantic information. For example, a user's frequent visits to university buildings may indicate they are a student or faculty member. Similarly, a vehicle's repeated trips between commercial and industrial zones might suggest it is a delivery vehicle. Extracting and utilizing such implicit information can enable models to better capture user/vehicle characteristics, transit habits, and movement preferences. This is particularly crucial in zero-shot scenarios, where statistical and spatiotemporal patterns alone are often insufficient for accurate predictions, and trajectory semantics play a vital role.
> > >
> > > However, existing small models, while achieving notable success in the next location prediction task, face two critical limitations that hinder their ability to capture the latent semantic information in trajectory data, thereby restricting further performance improvements:
> > > + **Loss of Semantic Context**: Small models require trajectory context information (e.g., spatiotemporal signals and POI category information) to be converted into numerical IDs or vectors. This transformation often results in the loss of the inherent meanings of trajectory context in the physical world.
> > > + **Limited Multimodal Integration**: Due to the architectural constraints of small neural networks, they rely heavily on large datasets to learn trajectory patterns. They lack the capability to analyze and understand the semantic information embedded in trajectories, especially when integrating non-numerical modalities such as textual data.
> > >
> > > In contrast, LLMs encapsulate rich information and rules about daily life through natural language [2][3] and possess powerful reasoning capabilities [3][4]. They have achieved remarkable success as enhancers, predictors, or agents in other spatiotemporal tasks [5]. Inspired by this, we integrate LLMs into a unified framework for the NextLocP task, addressing the deficiencies of small models:
> > > + **Enhanced Context Representation**: By providing LLMs with natural language descriptions of POI categories and other non-numerical information, we convey the functional attributes of locations more comprehensively, enriching trajectory context.
> > > +  **Task-Specific Prompting**: We design task-specific prefix prompts to clearly define the prediction task and data content, activating the LLM’s ability to analyze and interpret trajectory semantics, thereby enhancing prediction accuracy.
> > > These designs fully exploit the LLM's understanding of natural language and human behavior patterns. As a result, the model learns not only spatiotemporal patterns but also the deeper semantics within trajectories. In fully-supervised settings, our model outperforms small models, demonstrating the importance of trajectory semantics—i.e., the deeper meanings of human behavior patterns in the real world—in improving next location prediction. Moreover, in zero-shot scenarios, our model shows even more pronounced advantages, overcoming traditional models’ limitations in cross-city tasks. These experimental results highlight the benefits of integrating LLMs in the next location prediction task.

---

> > > > ### Author Response · Authors · 2024-11-24
> > > > **Further Response to Reviewer arN2(2/2)**
> > > >
> > > > To further validate the impact of LLM pre-trained knowledge, we conducted a comparative experiment. We replaced the LLM in our framework with a randomly initialized Transformer of the same parameter size and trained it on the same large-scale trajectory dataset. The results showed that this model performed significantly worse than NextLocLLM, particularly in zero-shot scenarios. This demonstrates the critical role of LLMs' pre-trained linguistic knowledge and reasoning capabilities in the next location prediction task.
> > > >
> > > > The comparison between LLaMA and GPT results is indeed perplexing. However, we believe this does not contradict our conclusion about the importance of LLM pre-trained knowledge in the next location prediction task. Both LLaMA and GPT consistently outperform small models, irrespective of size. However, the nuanced experimental results merit further exploration, which could have profound implications for integrating LLMs with spatiotemporal tasks in the future. We hypothesize that due to differences in the informational content of spatiotemporal and textual data, redundant pre-trained linguistic knowledge may offer limited benefits for spatiotemporal data predictions or even introduce issues similar to "hallucinations," adversely affecting performance. This could explain why the more complex and larger LLaMA model slightly underperforms GPT-2 in this context.
> > > >
> > > > Theoretically, LLaMA’s superior reasoning ability should confer stronger predictive power. Moving forward, identifying ways to mitigate the impact of redundant pre-trained knowledge while fully leveraging larger language models' reasoning capabilities is a promising avenue for research. We deeply appreciate the reviewers’ continued attention to this issue and look forward to exploring the potential and future directions of LLMs in spatiotemporal data mining. Thank you again for your valuable feedback!
> > > >
> > > > | Fully-Supervised | Hit@1   | Hit@5   | Hit@10  |
> > > > |------------------|---------|---------|---------|
> > > > | NextlocLLM       | 58.14%  | 97.14%  | 99.36%  |
> > > > | LLM->Transformer | 45.62%  | 83.25%  | 88.78%  |
> > > >
> > > > | Xi'an → Chengdu  | Hit@1   | Hit@5   | Hit@10  |
> > > > |------------------|---------|---------|---------|
> > > > | NextlocLLM       | 61.96%  | 96.89%  | 98.86%  |
> > > > | LLM->Transformer | 31.18%  | 51.65%  | 62.71%  |
> > > >
> > > > [1] Large Language Models for Next Point-of-Interest Recommendation
> > > >
> > > > [2]  Prompt-Based Time Series Forecasting: A New Task and Dataset
> > > >
> > > > [3] Large Language Models Are Zero-Shot Time Series Forecasters.
> > > >
> > > > [4] LSTPrompt: Large Language Models as Zero-Shot Time Series Forecastersby Long-Short-Term Prompt
> > > >
> > > > [5] What Can Large Language Models Tell Us about Time Series Analysis

---

### Official Review · Reviewer_za3U · 2024-11-02

**Soundness:** 3
**Presentation:** 3
**Contribution:** 2
**Rating:** 5
**Confidence:** 4

**Summary:**

The paper proposes a novel LLM-based framework, NextLocLLM, for next-location prediction. By integrating trajectory and POI data with a fine-tuned LLM, the framework aims to enhance prediction accuracy. The primary components include spatial coordinate encoding for better spatial representation, LLM-enhanced POI embeddings capturing functional location attributes, and a prediction retrieval module to provide top-k predictions.

**Strengths:**

• Innovative Framework: The use of LLM for next-location prediction is a key improvement, as it integrates semantic POI information, which is not fully leveraged by traditional methods.
•  Extensive Experiments: The authors present comprehensive experiments validating the model's performance across various datasets, demonstrating its robustness.
•  Clarity: The paper’s structure is well-organized, making the methodology clear and accessible.

**Weaknesses:**

• Innovation Concerns: The core contribution, POI information embedding, appears somewhat incremental since it aligns with previous works' embedding logic but integrates this with LLM-based representation.
•  Contribution of KD-tree: The KD-tree application is commonplace in traffic scenarios, particularly with GNN-based models, and thus may not qualify as a novel contribution.
•  Model Architecture Innovation: There appears to be limited innovation in the model structure. I would appreciate a discussion with the authors to ensure I am not overlooking any novel aspects.
•  Case Studies in Experiments: The inclusion of case studies could bolster the practical applicability of the framework by demonstrating real-world scenarios.
•  Minor Errors: Minor errors, such as in Equation 9, should be corrected for accuracy.

**Questions:**

Please refer to weakness. In addition, I hope the author can give an intuitive analysis of why the LLM-based model performs better. And what aspects of the design the author thinks are most effective, for example, the prompt prefix.

---

> ### Author Response · Authors · 2024-11-20
> **Response to Reviewer za3U (1/5)**
>
> >W1 Innovation Concerns: The core contribution, POI information embedding, appears somewhat incremental since it aligns with previous works' embedding logic but integrates this with LLM-based representation.
>
> In next location prediction, understanding POI information is essential for capturing the functional attributes of locations. This helps the model gain deeper insights into regional characteristics and better capture user behavior semantics. However, traditional methods typically compute POI embeddings directly from numerical vectors representing POI distributions. This approach overlooks the rich semantic information inherent in POI categories, limiting the model's ability to fully understand the functional attributes of locations.
>
> The limitations of traditional POI embedding methods become more pronounced in zero-shot cross-city tasks. These methods assume consistent POI taxonomy between the source and target cities [1][2]. However, in reality, there are significant differences in POI formats and taxonomy across cities. For example, in our study, the POI categories used in the datasets from Xi’an, Singapore, and Japan differ significantly, making it challenging to align POI categories strictly. This inconsistency hampers the generalization ability of traditional models in cross-city tasks, as POI embeddings derived solely from numerical vectors of POI distributions fail to resolve the semantic ambiguities caused by the inconsistency in POI taxonomy systems.
>
> To address this challenge, we propose an LLM-enhanced POI embedding method. This method leverages the natural language understanding capabilities of LLMs to transform the natural language descriptions of POI categories into semantic embedding vectors, thereby capturing the deep functional semantics of POI categories. Unlike traditional methods, this approach does not rely on fixed POI taxonomy but instead operates within a unified semantic space, enabling effective modeling even when POI definitions and classification systems differ across cities. Experimental results show that this method effectively resolves semantic ambiguities in zero-shot scenarios and exhibits significant advantages in cross-city tasks.
>
> [1] Jiping L, An L, Fuhao Z, et al. Hybrid Classification Method for Multi-source POIs Based on Semantic Analysis[J].
>
> [2]Jiang R, Song X, Fan Z, et al. Transfer urban human mobility via poi embedding over multiple cities[J]. ACM Transactions on Data Science, 2021, 2(1): 1-26.

---

> ### Author Response · Authors · 2024-11-20
> **Response to Reviewer za3U (2/5)**
>
> > W2 Contribution of KD-tree: The KD-tree application is commonplace in traffic scenarios, particularly with GNN-based models, and thus may not qualify as a novel contribution.
>
> We would like to clarify that our contribution lies beyond KD-tree. It lies in introducing a regression-based approach to replace traditional classification methods, providing a more effective solution for zero-shot next location prediction tasks. We will revise our manuscript to make this distinction clearer.
>
> Traditionally, next location prediction tasks are formulated as classification problems, where each possible location ID is treated as a class, and the model predicts a probability distribution over these classes to determine the next location. However, this approach has significant limitations in zero-shot scenarios. Since location ID systems differ completely between cities, classification-based methods fail to transfer effectively. Additionally, classification inherently relies on predefined discrete categories, making it incapable of predicting unseen locations.
>
> In contrast, NextLocLLM employs a regression-based approach that directly predicts the spatial coordinates of the next location instead of discrete class labels. This design effectively addresses the limitations of traditional classification methods. The regression approach leverages the continuous nature of spatial coordinates, enabling the prediction of unseen geographic locations and achieving better generalization by learning geographic spatial relationships. On the other hand, classification methods are restricted to predefined categories and fail to generalize in new scenarios. Furthermore, the regression approach eliminates the dependence on location IDs, removing the need for unified mappings across cities and significantly enhancing the applicability of the model in cross-city tasks. Additionally, while classification methods typically predict discrete grids or predefined locations, regression directly predicts precise latitude and longitude coordinates, offering superior geographic resolution.
>
> Within this framework, the KD-tree plays a supportive role to accelerate the post-processing of regression outputs by efficiently mapping the predicted coordinates to the nearest location in the candidate set through spatial search. This process is intended to enhance the inference efficiency  of the regression-based design.
>
> In summary, our innovation lies in introducing the regression-based design, which overcomes the core limitations of traditional classification methods and provides a more adaptive and generalizable solution for next location prediction in zero-shot and cross-city tasks. The KD-tree, meanwhile, serves as a tool to enhance the efficiency of this framework. We will clarify this distinction further in the revised manuscript. Thank you for your valuable feedback!
>
> > W3 Model Architecture Innovation: There appears to be limited innovation in the model structure. I would appreciate a discussion with the authors to ensure I am not overlooking any novel aspects.
>
>  To address your concerns, we further highlight the key innovations of NextLocLLM in terms of its model architecture.
>
> According to [1], the role of LLM in spatiotemporal data mining tasks can be categorized into three paradigms: LLM as enhancer, LLM as predictor, and LLM as agent. NextLocLLM is the first model in the field of next location prediction to integrate LLM as enhancer and LLM as predictor into a unified framework. This novel paradigm tightly combines the semantic understanding capabilities of pre-trained large language models with the geographic prediction task, offering a groundbreaking approach to next location prediction.
>
> By leveraging LLMs in this dual role, NextLocLLM not only addresses the limitations of traditional models in cross-city tasks but also establishes a new direction for the research and application of next location prediction. This integration allows the model to utilize both LLM-enhanced feature embeddings (e.g., POI semantics) and LLM-based regression capabilities for predicting spatial coordinates, enabling a more robust, adaptable, and generalizable solution.
>
> We deeply appreciate your insights and look forward to discussing future directions for next location prediction, particularly how the potential of LLMs can be further explored to enhance not only next location prediction but also other spatiotemporal tasks. Thank you again for your valuable feedback!
>
>
> [1] Jin M, Zhang Y, Chen W, et al. Position: What Can Large Language Models Tell Us about Time Series Analysis[C] ICML. 2024.

---

> ### Author Response · Authors · 2024-11-20
> **Response to Reviewer za3U (3/5)**
>
> > W4  Case Studies in Experiments: The inclusion of case studies could bolster the practical applicability of the framework by demonstrating real-world scenarios.
>
> To address your concerns, we have included a case study based on the Singapore dataset to showcase the real-world application of NextLocLLM and its significant advantages in zero-shot cross-city tasks. For details, please refer to Section P and Figure 8.
>
> In this case study, we evaluated the zero-shot capabilities of NextLocLLM trained on the Xi'an dataset by testing it on the Singapore dataset. We also compared these results with the fully-supervised setting, where the model was both trained and tested on the Singapore dataset. The Singapore dataset was chosen due to its extensive coverage of location IDs and its distinct POI classification system, which differs from that of Xi'an. This characteristic makes our case study more representative of real-world transfer scenarios, where different cities often use varying POI classification and definition systems, posing a challenge for models to generalize across cities.
>
> Specifically, we analyzed a user trajectory from the Singapore dataset with densely sampled points to provide an intuitive demonstration of the model's performance. The trajectory had an average sampling interval of 21 minutes. We compared the following approaches:
> + Zero-shot NextLocLLM (using coordinates)
> + Zero-shot NextLocLLM (using location IDs)
> + Fully-supervised NextLocLLM (using coordinates)
> + LLM-Mob prediction
>
> The experimental results clearly highlight the differences in performance across these methods: Zero-shot NextLocLLM (using coordinates) exhibited outstanding predictive capability, generating trajectories highly consistent with the ground truth. This demonstrates NextLocLLM's strong zero-shot generalization in cross-city tasks. In contrast, Zero-shot NextLocLLM (using location IDs) showed significant deviations from the ground truth. This result underscores the limitations of location ID-based models in cross-city tasks, as inconsistent location ID systems across cities fail to convey the semantic information of geographic spaces. LLM-Mob prediction consistently limited its forecasting to locations explicitly mentioned in the prompts, revealing that models purely based on prompts and location IDs struggle to comprehend cross-city spatial information or capture trajectory sequence patterns. This limitation significantly reduces the applicability of LLM-Mob for tasks involving unknown locations, particularly in zero-shot cross-city scenarios.Finally, Fully-supervised NextLocLLM (using coordinates) exhibits the superior performance in a supervised setting, achieving results only marginally better than the zero-shot setting. This further validates the robust generalization capability of NextLocLLM, which maintains near-supervised performance even without target city training data.
>
> Through this case study, we demonstrate the clear advantages of NextLocLLM in cross-city tasks, particularly its remarkable zero-shot generalization capabilities. Wel include a detailed analysis and visualizations of this case study in the revised manuscript.
>
> > W5  Minor Errors: Minor errors, such as in Equation 9, should be corrected for accuracy.
>
> Thank you for your valuable suggestions. We have revised our manuscript  and corrected the equation.

---

> ### Author Response · Authors · 2024-11-20
> **Response to Reviewer za3U (4/5)**
>
> > Q1 I hope the author can give an intuitive analysis of why the LLM-based model performs better.
>
> Thanks for your suggestions. Below, we provide a detailed analysis and supporting experiments to explain the strengths of LLMs and the rationale behind our design decisions.
>
> The primary advantage of integrating LLMs into this task lies in their ability to transcend the limitations of small models. Traditional small models typically rely on spatial and temporal patterns learned from large trajectory datasets to predict the next location. While effective in data-rich scenarios, these models often overlook the semantic nature of trajectories—the underlying human behavioral patterns reflected in the real world. In data-scarce or zero-shot scenarios, where statistical patterns alone are insufficient, trajectory semantics become critical. LLMs, with their powerful natural language understanding and reasoning capabilities, excel in capturing the inherent semantics of trajectories, even with limited data. This is the foundational motivation behind using LLMs as the core design of our framework.
>
> To maximize the utility of LLMs, we optimized our model to enhance its semantic understanding of trajectories. First, we employed prompt prefixes to clearly define the prediction task and data, coupled with natural language descriptions of POIs to convey locations' functional attributes. This design leverages the LLM’s ability to interpret natural language and human behavioral patterns, enabling it to go beyond learning spatial and temporal patterns and uncover the deeper semantic and logical relationships within trajectories. This capability is particularly beneficial for zero-shot tasks, where such semantic understanding significantly boosts performance.
>
> To validate our hypothesis, we conducted experiments where we replaced the LLM in our framework with a randomly initialized Transformer which shares the same amount of parameters, and trained it on Xi’an dataset. The results showed that this model performed significantly worse in zero-shot and fully-supervised tasks, with a particularly notable gap in zero-shot scenarios. This demonstrates the importance of LLM’s pre-trained knowledge and reasoning capabilities. The LLM effectively utilizes its internal semantic and logical information to complement the lack of statistical data. Even when the Transformer model was trained on a large amount of data, its performance still fell short of the LLM, further confirming the LLM’s unique strengths in zero-shot and transfer tasks.
>
> In fully supervised scenarios with abundant data, small models may hold certain advantages, such as computational efficiency and faster training times. However, in real-world applications, data scarcity and transfer tasks are common challenges. In these scenarios, the LLM’s ability to understand and reason about the semantics of trajectories gives it a clear edge.
>
> Looking ahead, we believe human trajectories inherently contain spatiotemporal intent and semantics. These deep characteristics offer promising directions for using LLMs in spatiotemporal data mining. For instance, LLMs can be further explored to uncover behavioral patterns, user preferences, and higher-level semantic relationships within trajectories, enabling them to address a broader range of tasks. We deeply appreciate the reviewer’s interest in this topic and look forward to further discussing the potential and future directions of LLMs in spatiotemporal data mining. Thank you for your valuable feedback!
>
> | Fully-Supervised | Hit@1   | Hit@5   | Hit@10  |
> |------------------|---------|---------|---------|
> | NextlocLLM       | 58.14%  | 97.14%  | 99.36%  |
> | LLM->Transformer | 45.62%  | 83.25%  | 88.78%  |
>
> | Xi'an → Chengdu  | Hit@1   | Hit@5   | Hit@10  |
> |------------------|---------|---------|---------|
> | NextlocLLM       | 61.96%  | 96.89%  | 98.86%  |
> | LLM->Transformer | 31.18%  | 51.65%  | 62.71%  |

---

> > ### Author Response · Authors · 2024-11-20
> > **Response to Reviewer za3U (5/5)**
> >
> > > Q1.1 what aspects of the design the author thinks are most effective, for example, the prompt prefix.
> >
> > For this question, we think it is the **LLM-enhanced POI embedding**. As shown in the ablation study results in Table 3, the impact of removing LLM-enhanced POI embeddings on model performance is greater than removing the prompt prefix. This underscores the critical role of LLM-enhanced POI embeddings in improving the model’s performance.
> >
> > We argue that one of the primary advantages of LLMs in the next location prediction task lies in their semantic understanding capabilities. Through the LLM-enhanced POI embedding mechanism, LLMs can better interpret the functional attributes of locations, enabling them to infer the human behavioral intentions embedded in trajectories. This is something that traditional small models struggle to achieve. Even when only LLM-enhanced POI embeddings are used, the model achieves high performance, indicating that the LLM’s ability to understand regional semantics contributes significantly to the overall results.
> >
> > Unlike traditional methods, which directly compute POI embeddings from numerical vectors representing POI distributions, LLM-enhanced POI embedding utilizes natural language descriptions of POI categories to generate semantic embeddings. This design leverages the natural language understanding capabilities of LLMs, enabling the model to capture the deeper functional semantics of POI categories. Consequently, it can better comprehend the human behavioral patterns and travel intentions underlying the trajectories, resulting in improved predictive performance.
> >
> >
> >
> > In summary, we believe that LLM-enhanced POI embedding is  the most effective aspects of our design. It not only fully leverages the semantic understanding capabilities of LLMs but also addresses the generalization challenges faced by traditional POI embedding methods in transfer scenarios.

---

### Official Review · Reviewer_xQuD · 2024-11-02

**Soundness:** 3
**Presentation:** 4
**Contribution:** 3
**Rating:** 6
**Confidence:** 5

**Summary:**

This article proposes a novel method for next location prediction based on fine-tuning large language models.

**Strengths:**

1. Well written, clear structure, easy to understand.

2. Proposed a series of key insights to enhance the adaptation of LLM in the next location prediction task.
   - Applying *Spatial Coordinate Normalization* to unleash LLM's cross-city generalization ability
   - Proposing *LLM-enhanced POI Embedding* to integrate diverse functional attributes of regions.
   - Predicting spatial coordinates as intermediate results, then using KD-tree retrieval for top-k locations, which is a **rather smart idea** that not only retain the generalization ability brought by coordinate prediction itself, but also is compatible with classic problem definition (Next **Location ID** prediction).

3. Conducted sufficient ablation study to demonstrate the importance of each module.

**Weaknesses:**

1. The **model parameters** and **configurations** are **completely lacking** in introduction.
     - **Basic training parameters**: learning rate, batch size, optimizer, etc.
     - **Key hyperparameters**: the dimension of $d_{llm}$, etc.
     - **Model Input**: Length of historical and current trajectories.
     - **LLM Tokenizer and LLM Token Embedding Layer**: Which models/methods are used for?

2. **Lack of sensitivity analysis on hyperparameters**.
   Although the authors conducted a complete ablation study demonstrating the key roles of each module, it is still necessary to conduct corresponding sensitivity analysis on key parameters to demonstrate the robustness of the model.

3. The overlooked issue of **computational efficiency**.
Inference efficiency of LLMs has always been a critical issue. The proposed model not only requires using *pre-trained LLM as the backbone network* but also needs to use *LLM encoding POI information and prompt prefix*. Therefore, I am very concerned about the computational efficiency issues of the proposed method, including training and inference time as well as resource utilization. I hope that the authors can report on this point and compare it with baselines. It is acceptable that the computational efficiency is not good for an ICLR paper, but it is better to discuss the drawback (if have)

**Questions:**

Q1. Why is the NextlocLLM model labeled as a **white box model** in Figure 1(c)? I did not find any evidence in this paper to support this claim.

Q2. In the conclusion section, the authors mentioned that the *geographical distance error remains a challenge*, and maybe further reducing the grid size can mitigate this? The current study used a grid size of 500m x 500m for location prediction. I would like to know how grid size affects accuracy.

---

> ### Author Response · Authors · 2024-11-20
> **Response to Reviewer xQuD (1/3)**
>
> > W1 The model parameters and configurations are completely lacking in introduction.
>
> To address your concerns, we present the detailed parameters and configurations for NextLocLLM as mentioned in your comments (see section K):
>
> **Basic Training Parameters**
> + Learning Rate: 0.0002
> + Batch Size: 128
> + Max Epochs: 30
> + Optimizer: Adam
>
> **Key Hyperparameters**
> + LLM Backbone: GPT-2
> + Spatial Coordinate Embedding Dimension: 128
> + Day & Hour Embedding Dimension: 16
> + Duration Embedding Dimension: 16
> + POI MLP Dimension: 1024
> + LLM Layers: 6
>
> **Model Input Configuration**
> + Historical Trajectory Length: 40 records, capturing long-term behavioral patterns of users.
> + Current Trajectory Length: 5 records, capturing short-term behavioral features.
>
> **LLM Tokenizer and Embedding Layer**
>
> We used the tokenizer and embedding layer consistent with the GPT-2 model to ensure compatibility and uniformity in input processing.
>
> **Experimental Hardware**
>
> The experiments were conducted on a set of Tesla V100-SXM2-32GB GPUs to support large-scale data training and inference.
>
> We include these detailed parameter and configuration descriptions in the revised manuscript to enhance the clarity of the model presentation and the reproducibility of the experiments. We greatly appreciate the reviewers’ valuable suggestions.
>
> >W2 Lack of sensitivity analysis on hyperparameters. Although the authors conducted a complete ablation study demonstrating the key roles of each module, it is still necessary to conduct corresponding sensitivity analysis on key parameters to demonstrate the robustness of the model.
>
> We thank the reviewers for their valuable suggestions. In response, we have further conducted a detailed sensitivity analysis of the key hyperparameters to validate the robustness of the model's performance (see Section Q for details). Below is a summary of the findings:
>
> Parameters like $d_{xy}$, $d_d$, $d_t$, and $d_{poi}$​ exhibit relatively stable effects on the model’s performance. For instance, increasing the spatial coordinate embedding dimension from 64 to 256 steadily enhances performance, but further increases show saturation, indicating that smaller dimensions effectively capture geographic information. Similarly, the time and date embedding dimension shows slight improvements as it grows from 8 to 32, with minimal gains beyond this range, reflecting the robustness of these features.
>
> On the other hand, parameters such as the historical trajectory length ($M$) and the number of LLM layers ($N_{layers​}$) have a more pronounced impact. Increasing the historical trajectory length improves the model’s ability to capture past travel patterns. Moreover, performance improves significantly as the number of layers increases, indicating that deeper architectures enhance the model’s representational capacity. However, beyond 6 layers, the performance stabilizes. Considering the trade-off between efficiency and performance, we chose to use a 6-layer GPT configuration.
>
> This sensitivity analysis underscores the robustness of the model to most parameter settings while identifying key parameters with greater influence on performance. These results and analyses are included in the revised manuscript to provide a more comprehensive evaluation. Thank you again for your valuable feedback!

---

> > ### Author Response · Authors · 2024-11-20
> > **Response to Reviewer xQuD (2/3)**
> >
> > > W3 The overlooked issue of computational efficiency. Inference efficiency of LLMs has always been a critical issue. The proposed model not only requires using pre-trained LLM as the backbone network but also needs to use LLM encoding POI information and prompt prefix. Therefore, I am very concerned about the computational efficiency issues of the proposed method, including training and inference time as well as resource utilization. I hope that the authors can report on this point and compare it with baselines. It is acceptable that the computational efficiency is not good for an ICLR paper, but it is better to discuss the drawback (if have)
> >
> > To address your concerns regarding model efficiency, we have conducted an in-depth analysis and supplementary experiments focusing on training and inference times (see Section R).
> >
> > For training, the per-iteration training time of NextLocLLM is 0.935 seconds, which is higher compared to GRU (0.131 seconds), DeepMove (0.155 seconds), and c-MHSA (0.199 seconds). However, during training, NextLocLLM reduces its computational burden by freezing most of the LLM backbone parameters and only tuning a small number of parameters. This design reduces the number of trainable parameters, thereby accelerates model convergence. For instance, on the Xi'an dataset, NextLocLLM converges in approximately 15 epochs. As a result, the total training time is tolerable, and the substantial performance improvement achieved by the model justifies the higher computational cost.
> >
> > In terms of inference time among LLM-based models, NextLocLLM demonstrates a clear advantage. For example, on the Xi'an dataset, its inference time is 229.2 seconds, which is significantly lower than LLM-Mob’s 9,522 seconds. This efficiency gain can be attributed to two key factors. First, NextLocLLM supports batch inference, allowing it to predict multiple trajectories simultaneously, in contrast to LLM-Mob and ZS-NL, which rely on prompt engineering and process trajectories sequentially. This parallelized design greatly reduces the total inference time. Second, the POI functional attributes for each location remain consistent across trajectory records. In NextLocLLM, the LLM-enhanced POI embeddings are precomputed once during data loading and stored for reuse, eliminating the need for repeated calculations during inference and substantially reducing the computational overhead.
> >
> > We acknowledge that there is room for improvement in NextLocLLM’s computational efficiency. In future work, we plan to further optimize the model by exploring techniques such as quantization and pruning to reduce computational overhead through optimized model weights and structures. Additionally, we aim to investigate lightweight model designs through distillation methods to further reduce the model size and enhance real-time performance. Thank you for your suggestion. It will be our honor and pleasure to discuss with you further about this potential direction!
> >
> > | Model       | Training Time (s/iter) |
> > |-------------|-------------------------|
> > | GRU         | 0.131                   |
> > | Deepmove    | 0.155                   |
> > | c-MHSA      | 0.199                   |
> > | NextlocLLM  | 0.935                   |
> >
> > | Model       | Inference Time (s) [whole Xi'an testing set] |
> > |-------------|---------------------------------------------|
> > | NextlocLLM  | 229.2                                       |
> > | LLM-Mob     | 9522                                        |
> >
> >
> > >Q1 Why is the NextlocLLM model labeled as a white box model in Figure 1(c)? I did not find any evidence in this paper to support this claim.
> >
> > Thank you for your question. We referenced several related studies that explicitly define the term “white-box LLM,” providing the basis for our labeling. Specifically, works such as [1],[2] and [3] define “white-box LLMs” as models where their internal parameters are directly accessible. This contrasts with “black-box LLMs,” which are typically accessed through APIs or encapsulated interfaces that do not allow users to view or modify their internal parameters.
> >
> > Following this definition, NextLocLLM qualifies as a “white-box LLM” because its parameters are fully accessible and can be efficiently fine-tuned. To prevent potential misunderstandings among readers, we include a detailed explanation of the “white-box LLM” concept in the revised manuscript and add an explanation in the caption of Figure 1 to clarify the distinction between white-box and black-box models.
> >
> > [1] Jin M, Zhang Y, Chen W, et al. Position: What Can Large Language Models Tell Us about Time Series Analysis[C] ICML. 2024.
> >
> > [2] Chen Z, Chen P Y, Buet-Golfouse F. Online personalizing white-box llms generation with neural bandits[C]//Proceedings of the 5th ACM International Conference on AI in Finance. 2024: 711-718.
> >
> > [3]Hong J, Tu Q, Chen C, et al. Cyclealign: Iterative distillation from black-box llm to white-box models for better human alignment[J].  2023.

---

> > > ### Author Response · Authors · 2024-11-20
> > > **Response to Reviewer xQuD (3/3)**
> > >
> > > > Q2. In the conclusion section, the authors mentioned that the geographical distance error remains a challenge, and maybe further reducing the grid size can mitigate this? The current study used a grid size of 500m x 500m for location prediction. I would like to know how grid size affects accuracy
> > >
> > > To answer your question, we have conducted additional experiments on Xi’an dataset to evaluate the performance of NextLocLLM and C-MHSA under different grid sizes (500m×500m, 250m×250m, and 50m×50m) and analyzed how grid resolution affects the models' performance.
> > >
> > >
> > > The experimental results show that for C-MHSA, finer grid resolutions lead to a noticeable decline in performance. When the grid size is reduced to 50m×50m, the number of candidate locations increases substantially, resulting in significantly higher computational overhead and eventually causing an out-of-memory (OOM) error. In contrast, NextLocLLM, which directly predicts coordinates without relying on grid partitions, successfully completes predictions even under fine-grained grid settings. While its performance declines at higher resolutions, it still outperforms C-MHSA overall, demonstrating robust adaptability to changes in spatial resolution.
> > >
> > > However, this experiment also highlights some areas for potential improvement in NextLocLLM when handling high-resolution prediction tasks. For instance, under extremely fine grid settings (e.g., 50m×50m), the model’s performance still has room for enhancement, suggesting that its ability to model coordinate precision could be further improved.
> > >
> > > We are deeply grateful for the reviewers’ suggestion, as this issue points to promising future research directions. We plan to explore more efficient optimization strategies, such as incorporating dynamic grid partitioning or hierarchical mechanisms, to further improve the model's performance in high-resolution scenarios. We look forward to engaging in deeper discussions with you on this topic in the future!
> > >
> > > | Grid       | C-MHSA               |                    |                    | NextLocLLM          |                    |                    |
> > > |------------|-----------------------|--------------------|--------------------|----------------------|--------------------|--------------------|
> > > |            | Hit@1                | Hit@5             | Hit@10             | Hit@1               | Hit@5             | Hit@10             |
> > > | 500*500    | 50.32%               | 92.43%            | 95.38%             | 58.14%              | 97.14%            | 99.36%             |
> > > | 250*250    | 40.63%               | 82.26%            | 86.77%             | 50.99%              | 88.42%            | 94.77%             |
> > > | 50m*50m    | OOM                  | OOM                | OOM                | 36.87%              | 62.95%            | 72.47%             |
> > >
> > >
> > > .

---

> > > > ### Comment · Reviewer_xQuD · 2024-11-24
> > > > **Response from xQuD**
> > > >
> > > > Thank you to the author for the detailed responses. My questions have all been answered. I also carefully read the comments from other reviewers, especially reviewer arN2, and I agree with his views. Regarding what makes GPT-2 better than Llama2 or Llama3, introducing LoRA and POI is not as effective as just introducing POI; currently, these are merely subjective interpretations of the experimental results. Therefore, I will maintain my score. Thank you.

---

> > > > > ### Author Response · Authors · 2024-11-25
> > > > > **Further Response to Reviewer xQuD**
> > > > >
> > > > > Thank you for your feedback. We are pleased to find that we have addressed all the questions you raised.  Regarding why GPT-2 outperforms Llama2 and Llama3 as the backbone, and why fine-tuning the attention and fully connected layers of LLMs using LoRA yields worse results than freezing these parameters, both phenomena indeed seem somewhat "counterintuitive." Such occurrences are not uncommon in current research on spatiotemporal data mining [1][2][3][4][5][6]. However, to the best of our knowledge, there is no systematic theoretical study specifically exploring the knowledge boundaries or "hallucinations" of large models in spatiotemporal, geographic, or time-series tasks.
> > > > >
> > > > > We believe these are fascinating and worthwhile topics for further exploration, particularly in analyzing the behavior, knowledge boundaries, hallucination issues, and task adaptability of large models in non-natural language tasks. However, this falls beyond the scope of the current paper. In the future, we plan to conduct more experiments and theoretical studies to delve deeper into these phenomena and uncover the underlying mechanisms.
> > > > >
> > > > > Once again, thank you for your attention to this issue. We look forward to the opportunity for deeper discussions and collaborations with you in this area in the future!
> > > > >
> > > > > [1] Tempo: Prompt-Based Generative Pre-Trained Transformer for Time Series Forecasting
> > > > >
> > > > > [2] AutoTimes: Autoregressive Time Series Forecasters via Large Language Model
> > > > >
> > > > > [3] Enhancing Graph Neural Networks with Limited Labeled Data by Actively Distilling Knowledge from Large Language Model
> > > > >
> > > > > [4] NuwaTS: Mending Every Incomplete Time Series
> > > > >
> > > > > [5] Where would i go next? large language models as human mobility predictors
> > > > >
> > > > > [6] Spatial-Temporal Large Language Model for Traffic Prediction

---

### Official Review · Reviewer_7tUX · 2024-11-03

**Soundness:** 2
**Presentation:** 3
**Contribution:** 3
**Rating:** 5
**Confidence:** 4

**Summary:**

The paper presents NextLocLLM, a model designed to predict the next location by enhancing the extraction of spatial relationships and improving generalizability across different cities using a large language model (LLM) approach. First, NextLocLLM employs normalized spatial coordinates to represent discrete locations, accurately modeling spatial relationships and bypassing location ID inconsistencies across cities. Second, the model integrates LLM-enhanced POI category information, which captures functional attributes of locations more effectively. Finally, NextLocLLM leverages a KD-tree to convert output coordinates into the top-k most likely predicted locations, thereby incorporating neighborhood spatial relationships into the prediction process.

**Strengths:**

S1. Unlike other methods that rely solely on location IDs for prompt design, NextLocLLM focuses on spatial relationships and semantic embeddings derived from natural language descriptions of POI categories. By using spatial coordinates, the model gains a deeper understanding of spatial relationships between locations, enhancing transferability and generalization across diverse urban environments.
S2. The authors conduct extensive experiments with various models and baselines. These experiments are comprehensive, covering four datasets and demonstrating strong performance in both supervised and zero-shot settings.
S3. The paper is well-written and easy to follow.

**Weaknesses:**

W1.  Ablation studies to assess whether using spatial coordinates provides a clear advantage over location IDs is missing.
2. The paper does not include a detailed description of raw trajectory processing. For instance, there is no clarification on determining the length of historical and current trajectories or whether noisy points in the raw trajectory data are filtered out.
3. The paper could benefit from further analysis on which specific design modules contribute to the zero-shot capability of NextLocLLM and why it outperforms other LLM-based methods.

**Questions:**

Q1. How is the frequency of each POI category within a location calculated?
Q2. In fully-supervised scenarios, such as for training methods like DeepMove, is the same data used as with NextLocLLM? Additionally, are the POI category and stay duration included as training data for DeepMove?

---

> ### Author Response · Authors · 2024-11-20
> **Response to Reviewer 7tUX (1/3)**
>
> > W1 Ablation studies to assess whether using spatial coordinates provides a clear advantage over location IDs is missing.
>
> We sincerely appreciate the reviewers' suggestions regarding the ablation study in our paper. To investigate whether using spatial coordinates provides advantages over location IDs, we conducted comparative experiments (see Section O and the table below), where location IDs were directly used as inputs and outputs for the model. These results were then compared to the settings where spatial coordinates were employed.
>
> The experimental results reveal that in the fully supervised scenario, the model utilizing spatial coordinates outperforms the one using location IDs across all metrics on both the Xi'an and SG datasets. This indicates that spatial coordinates better capture geographic relationships, thereby improving prediction performance in fully supervised settings. The advantage of spatial coordinates becomes even more pronounced in zero-shot scenarios. Models relying on location IDs perform poorly in zero-shot settings, with all metrics nearing 0%. This result further highlights the limitations of using location IDs, particularly in generalizing across cities. The discrepancy in performance indicates that location IDs fail to capture the continuity of geographic space, and the location IDs vary significantly between cities, which constrains the model's ability to generalize across urban regions.
>
> To illustrate the superiority of spatial coordinates over location IDs more intuitively, we also present a case study (see Section P). This case study demonstrates that in zero-shot scenarios, NextlocLLM using spatial coordinates generates predictions that are significantly closer to the actual trajectories. In contrast, the predictions from the location ID-based model deviate substantially from the true trajectories.
>
> Both experiments demonstrate the significant advantages of spatial coordinates over location IDs. Spatial coordinates not only enhance the model's ability to capture geographic relationships but also substantially improve its generalization performance in cross-city scenarios. We include a detailed analysis and discussion of these experimental results in our revised manuscript.
>
> | Method                     | Coordinate Hit@1 | Coordinate Hit@5 | Coordinate Hit@10 | Location Hit@1 | Location Hit@5 | Location Hit@10 |
> |----------------------------|------------------|-------------------|-------------------|----------------|----------------|-----------------|
> | Xi'an (fully supervised)   | 58.14%           | 97.14%            | 99.36%            | 54.57%         | 85.11%         | 87.88%          |
> | Singapore (fully supervised) | 5.442%          | 17.14%            | 22.36%            | 2.061%         | 11.42%         | 15.54%          |
> | Xi'an → Singapore          | 5.026%           | 16.55%            | 19.46%            | 0%             | 0%             | 0%              |
> | Xi'an → Chengdu            | 61.96%           | 96.89%            | 98.86%            | 0%             | 0%             | 0.03%           |
> | Singapore → Xi'an          | 3.85%            | 9.55%             | 19.86%            | 0%             | 0%             | 0.05%           |
>
>
> > W2 The paper does not include a detailed description of raw trajectory processing. For instance, there is no clarification on determining the length of historical and current trajectories or whether noisy points in the raw trajectory data are filtered out.
>
> Thanks for your suggestion. Regarding the detailed description of data preprocessing, we  add comprehensive rules and procedures in Section I, along with Figure 6 as an illustrative diagram to visually demonstrate the effects of noise filtering and data cleaning. Below is a brief summary of the key steps:
>
> + Noise Filtering: Using the trackintel library to generate staypoints, we applied rules based on time intervals (60 minutes), distance thresholds (200 meters), and minimum dwell time (10 minutes or 1 minute for the Singapore dataset) to effectively eliminate noisy data.
> + Ensuring Trajectory Completeness: We filtered out trajectories with fewer than 15 days of coverage, removing short-term noise trajectories and ensuring the model learns long-term behavioral patterns.
> + Spatial Grid Partitioning and POI Mapping: The urban area was divided into grids of 500m × 500m, and the number of each type of POI in each grid was calculated. This process introduced functional characteristics of regions into the trajectory data, capturing semantic and behavioral patterns.
> + Optimizing Historical Trajectory Length: Based on hyperparameter sensitivity analysis, we determined that setting the historical trajectory length to 30–40 records as a default configuration ensures the model's stability and performance.

---

> ### Author Response · Authors · 2024-11-20
> **Response to Reviewer 7tUX (2/3)**
>
> > W3 The paper could benefit from further analysis on which specific design modules contribute to the zero-shot capability of NextLocLLM and why it outperforms other LLM-based methods.
>
> Thank you for your valuable feedback! To provide a  comprehensive explanation of the model’s performance in zero-shot scenarios, we conducted additional experiments and provided detailed discussions in Section S, focusing on the contributions of key design components to zero-shot capability, particularly spatial coordinate inputs and prompt prefix. The analysis is summarized as follows:
>
> + **The Role of Spatial Coordinate Inputs**: Experimental results demonstrate that using spatial coordinates significantly improves the model’s performance in zero-shot scenarios compared to location IDs, validating spatial coordinates as a universal representation of geographic information.
> + **The Importance of prompt prefix**: Removing prompt prefix leads to a significant performance drop, highlighting their critical role in helping the model understand the data characteristics and task objectives.
>
>
> Additionally, regarding why NextLocLLM outperforms other LLM-based models in zero-shot scenarios, we suggest that this advantage is closely tied to the use of spatial coordinates as inputs. Pre-trained LLMs are not inherently proficient in understanding spatial relationships between coordinates [1]. Consequently, previous LLM models relying on prompt engineering often used location IDs to represent spatial information. However, as observed in our case study (see section P), prompt-based models using location IDs can only predict locations explicitly seen in the prompt, failing entirely to generalize to unseen locations. This limitation severely impacts their applicability and accuracy. In contrast, NextLocLLM’s integration of spatial coordinates allows the model to generalize to unseen geographic regions, resulting in significant performance improvements in zero-shot tasks.
>
> | Xi'an → Chengdu | Hit@1  | Hit@5  | Hit@10 |
> |------------------|---------|---------|---------|
> | Location         | 0%      | 0%      | 0.03%   |
> | Coordinate       | 61.96%  | 96.89%  | 98.86%  |
> | No Prompt Prefix | 45.84%  | 86.11%  | 93.84%  |
>
> [1] Manvi R, Khanna S, Mai G, et al. Geollm: Extracting geospatial knowledge from large language models[J]. ICLR 2024
>
> > Q1. How is the frequency of each POI category within a location calculated?
>
> Similar to [2], the frequency of POI categories at each location was calculated through the following steps. First, we mapped different types of POIs to their corresponding geographic grids. This allowed us to identify the specific POI types present within each grid. Next, we counted the frequency of each POI category in every grid, accurately capturing the distribution of POI types across different regions.
>
> To better represent the functional attributes of each area, we aggregated fine-grained POI categories into higher-level functional groups. This aggregation was necessary because individual fine-grained POI categories (e.g., Restaurants, Cafés) often fail to comprehensively reflect the overall functionality of an area. By combining similar or related POI categories, the aggregated functional groups provide a semantically richer and more intuitive representation. For example, a grid containing multiple restaurants and cafés can be classified under the broader functional category of Dining, which directly reflects the primary use of the area without focusing on the specifics of individual POI types. This aggregation process was based on the classification rules outlined in Tables 6 and 7.
>
> Through these steps, we constructed detailed POI frequency distributions for each location, providing semantically enriched functional attributes that are essential for supporting the model’s learning process. We include a detailed explanation of this POI frequency calculation and aggregation process in the revised manuscript. Additionally, this part of the code has already been included in our open-source repository for researchers to reference and use.
>
> [2]Luo G, Ye J, Wang J, et al. Urban functional zone classification based on POI data and machine learning[J]. Sustainability, 2023, 15(5): 4631.

---

> > ### Author Response · Authors · 2024-11-20
> > **Response to Reviewer 7tUX (3/3)**
> >
> > >  Q2. In fully-supervised scenarios, such as for training methods like DeepMove, is the same data used as with NextLocLLM? Additionally, are the POI category and stay duration included as training data for DeepMove?
> >
> > In our experiments, to ensure fairness, both NextLocLLM and DeepMove were trained on the same dataset. However, due to differences in model architecture and feature requirements, we adjusted the usage of data features accordingly.
> >
> > First, regarding the spatial representation of trajectories, traditional models like DeepMove typically use location IDs to represent spatial information. In contrast, NextLocLLM utilizes corresponding spatial coordinates to represent positions. Second, in NextLocLLM, we introduced POI categories and stay durations as additional features. These features provide richer information about the functional attributes of locations and user behavior patterns. However, DeepMove’s original architecture was not designed to process such additional features. We chose to keep the original architecture of DeepMove and did not include POI categories or stay durations in its training.
> >
> > Moreover, as shown in the ablation study in Table 4, NextLocLLM outperforms DeepMove even when POI categories and stay durations are excluded. This demonstrates that the architectural design of NextLocLLM and its semantic modeling capabilities based on large language models provide strong generalization and predictive performance, even without relying on additional features.

---

> > > ### Comment · Reviewer_7tUX · 2024-11-25
> > >
> > > I appreciate authors' efforst on answering my questions. Most of my concerns have been addressed. However, as also pointed out by other reviewers, the novelty of this work is limited, which mainly focuses on POI embedding. Therefore, I still believe my rating is appropriate.

---

> ### Author Response · Authors · 2024-11-25
> **Further Response to Reviewer 7tUX**
>
> Thank you for your feedback. We are pleased to know that we have addressed most of your concerns. Regarding your doubts about the novelty of this work, we would like to take this opportunity to summarize our key contributions beyond the LLM-enhanced POI embedding:
>
> 1. **Replacing Location IDs with Spatial Coordinates for Location Representation**
>
> Existing methods typically use numeric identifiers (location IDs) to represent locations. However, these discrete IDs fail to capture geographical relationships between locations and cannot generalize across cities due to the significant differences in location ID systems between cities. We propose using spatial coordinates instead of discrete IDs to represent geographical information.
>
> Our experiments—including comparisons between geographic coordinates and location IDs, ablation studies on zero-shot prediction performance, and case studies—demonstrate the clear advantages of spatial coordinates. These benefits are evident in both fully-supervised tasks and zero-shot tasks, where spatial coordinates significantly improve performance.
>
> 2. **A Regression-Based Approach for Zero-Shot Next Location Prediction**
>
> We introduce a regression-based approach that directly predicts the spatial coordinates of the next location, replacing the traditional classification-based design and offering a more effective solution for zero-shot tasks. Conventionally, next location prediction is treated as a classification problem where each possible location ID is considered a class, and the model predicts the probability distribution over these classes to determine the next location. However, this approach faces significant limitations in zero-shot scenarios:
> + *Limited transferability*: Classification models are bound to city-specific location ID systems, which vary widely and hinder effective cross-city generalization.
> + *Lack of generalization*: Classification relies on predefined classes and cannot predict unseen locations.
> + *Low geographical resolution*: Classification methods typically predict discrete grids or locations, whereas regression provides precise latitude and longitude predictions.
>
> In contrast, our regression-based method directly predicts the spatial coordinates of the next location, eliminating dependence on location IDs and significantly improving the applicability of the model to zero-shot cross-city tasks.
>
> 3.  **First Integration of LLM as Both Enhancer and Predictor in a Unified Framework for Next Location Prediction**
>
> Large language models (LLMs) encapsulate extensive knowledge and rules about daily life [1][2], exhibit strong reasoning abilities [2][3], and have achieved remarkable success in spatiotemporal tasks when used as data enhancers, predictors, or agents [4]. Inspired by this, we are the first to integrate LLMs as both enhancers and predictors within a unified framework for next location prediction. This novel paradigm tightly combines the semantic understanding capabilities of pretrained LLMs with geographic prediction tasks, offering a groundbreaking solution for next location prediction.
>
> By leveraging LLMs in dual roles, NextLocLLM not only overcomes the limitations of traditional models in cross-city tasks but also opens new avenues for research and applications in next location prediction. This integration enables the model to utilize both LLM-enhanced feature embeddings (e.g., POI semantics) and LLM-based regression capabilities for spatial coordinate prediction, resulting in a more robust, adaptive, and generalizable solution.
>
> We believe the above three contributions, together with the LLM-enhanced POI embedding, constitute the core innovations of this work. These designs address the limitations of traditional approaches while introducing a novel paradigm for next location prediction research.
>
> Thank you again for your valuable feedback! We look forward to future opportunities for further discussion and collaboration!
>
> [1] Prompt-Based Time Series Forecasting: A New Task and Dataset
>
> [2] Large Language Models Are Zero-Shot Time Series Forecasters
>
> [3] LSTPrompt: Large Language Models as Zero-Shot Time Series Forecasters by Long-Short-Term Prompt
>
> [4] What Can Large Language Models Tell Us about Time Series Analysis

---

### Author Response · Authors · 2024-11-20
**General Response To All Reviewers**

We sincerely thank all reviewers for their insightful and constructive feedback, which significantly contributed to improving the quality of our paper.  We are pleased to see the reviewers recognize our work's clarity and readability (Reviewer 7tUX, Reviewer xQuD, Reviewer za3U). We also appreciate their acknowledgement of our model's strong transferability and generalization capabilities (Reviewer 7tUX, Reviewer arN2). Furthermore, we are delighted that the reviewers find our experimental evaluations extensive and sufficient (Reviewer 7tUX, Reviewer xQuD, Reviewer za3U) and view our proposed idea as smart and impactful (Reviewer xQuD, Reviewer za3U, Reviewer arN2).

The major revisions and new experiments conducted in response to the reviewers' comments are summarized below:
+ We include new experiments comparing the fully-supervised and zero-shot performance of our model when using spatial coordinates versus location IDs as inputs.
+ We add a case study to highlight the practical advantages of NextLocLLM in cross-city tasks, demonstrating its real-world applicability.
+ We append a comprehensive list of model parameters and configurations to ensure improved reproducibility.
+ We conduct a detailed sensitivity analysis of the key hyperparameters to evaluate the robustness of the model's performance.
+ We conduct additional experiments to analyze training and inference times.
+ We conduct experiments where the LLM in our framework was replaced with a randomly initialized Transformer model, showcasing the critical role of the LLM in our approach.
+ We conduct experiments to analyze how varying grid resolutions affect the model's performance
+ We included new ablation studies on various components of the framework, focusing on their contributions to zero-shot performance.
+ We answer the concerns raised by reviewers.
+ We revised our paper to account for all changes, and the revisions are highlighted in blue color in the revised manuscript for your convenience.

We hope that our answers and revision of our paper will address your concerns.

---

### Note · Program_Chairs · 2024-12-02
**Submission Desk Rejected by Program Chairs**

The revised PDF contains the author names on the first page, breaking double blind review.